# Identification of candidate mitochondrial inheritance determinants using the mammalian cell-free system

**Dalen Zuidema[1], Alexis Jones[1], Won-Hee Song[1], Michal Zigo[1], Peter Sutovsky[1,2]***

[1]Division of Animal Sciences, University of Missouri, Columbia, United States; [2]Department of Obstetrics, Gynecology and Women's Health, University of Missouri, Columbia, United States

*For correspondence:
sutovskyp@missouri.edu

**Competing interest:** The authors declare that no competing interests exist.

**Abstract** The degradation of sperm-borne mitochondria after fertilization is a conserved event. This process known as post-fertilization sperm mitophagy, ensures exclusively maternal inheritance of the mitochondria-harbored mitochondrial DNA genome. This mitochondrial degradation is in part carried out by the ubiquitin-proteasome system. In mammals, ubiquitin-binding pro-autophagic receptors such as SQSTM1 and GABARAP have also been shown to contribute to sperm mitophagy. These systems work in concert to ensure the timely degradation of the sperm-borne mitochondria after fertilization. We hypothesize that other receptors, cofactors, and substrates are involved in post-fertilization mitophagy. Mass spectrometry was used in conjunction with a porcine cell-free system to identify other autophagic cofactors involved in post-fertilization sperm mitophagy. This porcine cell-free system is able to recapitulate early fertilization proteomic interactions. Altogether, 185 proteins were identified as statistically different between control and cell-free-treated spermatozoa. Six of these proteins were further investigated, including MVP, PSMG2, PSMA3, FUNDC2, SAMM50, and BAG5. These proteins were phenotyped using porcine in vitro fertilization, cell imaging, proteomics, and the porcine cell-free system. The present data confirms the involvement of known mitophagy determinants in the regulation of mitochondrial inheritance and provides a master list of candidate mitophagy co-factors to validate in the future hypothesis-driven studies.

## eLife assessment

This **important** work reports the identification of a list of proteins that may participate in the clearance of paternal mitochondria during fertilization, which is known as essential for normal fertilization and embryonic and fetal development. The main method used is state-of-the-art and the supporting data are **solid**. This work will be of interest to developmental and reproductive biologists working on fertilization.

## Introduction

Mitochondria are specialized cellular organelles which serve as the powerhouses of the cell. Mitochondria have been designated as such because of the large amounts of adenosine 5'-triphosphate (ATP) which they generate through both the TCA cycle as well as the electron transport chain. Additionally, mitochondria play roles in various cellular signaling pathways, serve as a calcium ion storage structure, help regulate cellular metabolism (*McBride et al., 2006*), and can signal for apoptosis (*Hajnóczky et al., 2006*). Mitochondria are also unique among cellular organelles because they house their own genome distinct from that of the nuclear DNA. This DNA is referred to as mitochondrial or mtDNA. The mtDNA encodes for 13 proteins which are used within the electron transport chain, as well as 22

tRNAs, and 2 rRNAs (*Anderson et al., 1981*). Mitochondria and the mtDNA which they harbor are almost exclusively inherited from the maternal lineage in most animal species, including humans. This pattern of maternal inheritance results in a single haplotype of the mtDNA being present in offspring.

This pattern of maternal inheritance results in most individual animals inheriting one mitochondrial genome from their mothers. Although naturally occurring maternally derived heteroplasmy (the presence of two distinct populations of mtDNA haplotypes) caused by mutations of the mitochondrial genome is observed (*McFarland et al., 2007*), heteroplasmy caused by the inheritance of the father's mtDNA is very rare (*Luo et al., 2018*; *Schwartz and Vissing, 2002*). Furthermore, in laboratory settings, researchers have been able to cause heteroplasmy in mice (*Sharpley et al., 2012*). These heteroplasmic mice were both mentally and physiologically insufficient when compared to their homoplasmic counterparts. The nematode roundworm *Caenorhabditis elegans* has also been utilized in this area of research. These worms were given a specific mtDNA deletion which caused the entire population to become heteroplasmic. It was found that embryonic lethality was 23-fold higher; they had reduced metabolic rates, and the males experienced reduced sperm mobility in this population of worms when compared to homoplasmic counterparts (*Liau et al., 2007*). Thus, in both laboratory animal populations, heteroplasmy was shown to be detrimental. It has also been observed that this mtDNA maternal inheritance pattern has been naturally violated in some human offspring (*Luo et al., 2018*; *Schwartz and Vissing, 2002*; *Slone et al., 2020*). Several multigenerational families have been identified with various forms of paternal mtDNA leakage. In the case of these studies, the heteroplasmy was identified because one of the family members had a mitochondrial disease. Although some of the relatives seemed to carry benign levels of paternal heteroplasmy, other family members were not so fortunate. Again, showing that when paternal mitochondria are not eliminated, it appears to eventually result in reduced health and fitness outcomes.

The paternal, sperm-borne mitochondria are located on the midpiece of the sperm tail, within a structure known as the mitochondrial sheath. This sheath organizes mitochondria into a compact helical configuration and in the livestock mammalian species, this sheath contains 50–75 mitochondria (*Hecht et al., 1984*). This mitochondrial sheath structure is quickly degraded from the fertilizing spermatozoa upon entry into the oocyte cytoplasm in a process known as post-fertilization sperm mitophagy. This mitophagic process is a nuanced autophagic and proteasome-dependent degradation of the mitochondria. It has been shown that ubiquitin associates with the sperm mitochondria in primate, ruminant, and rodent oocyte cytoplasm (*Sutovsky et al., 1999*). It has also been demonstrated that the ubiquitination of sperm mitochondria takes place in the porcine zygote, where the effect of proteasomal inhibitors on sperm mitophagy was described for the first time and resulted in implicating the ubiquitin-proteasome system as a pivotal part of this mitophagic process (*Sutovsky et al., 2004*; *Sutovsky et al., 2003*). More recently, studies using *Caenorhabditis elegans* and *Drosophila melanogaster* revisited the concept of ubiquitin-dependent post-fertilization sperm mitophagy, with a focus on the autophagic branch of this complex pathway (*Al Rawi et al., 2011*; *Sato and Sato, 2011*; *Zhou et al., 2011*; *Politi et al., 2014*). Much of the data from this research has focused on a handful of autophagic proteins. These proteins included sequestosome 1 (SQSTM1), GABA type A receptor-associated protein (GABARAP), microtubule-associated protein 1 light chain 3α (LC3), and valosin-containing protein (VCP). Further research has shown synergistic efforts between SQSTM1, GABARAP, and VCP in porcine sperm mitophagy (*Song et al., 2016*). To build upon this research, a better understanding of the potential co-factors, substrates, other proteins, and other pathways involved in post-fertilization sperm mitophagy is necessary.

The porcine cell-free system used in the present study is designed to recapitulate early fertilization proteomic interactions which would take place upon the incorporation of the spermatozoa into the ooplasm. It is a powerful tool that was used in this study (*Song and Sutovsky, 2018*), adapted from a similar amphibian system utilizing the eggs of an African clawed frog, *Xenopus laevis* (*Sutovsky et al., 1998*; *Miyamoto et al., 2009*; *Miyamoto et al., 2007*). Our porcine system utilizes small porcine oocytes (100 μm diameter) matured in vitro, instead of very large, easy-to-harvest frog eggs (1.2 mm diameter). Compared to other mammalian models, porcine oocytes are relatively easy to collect and mature (to fertilization-ready metaphase II stage) in large quantities. Furthermore, the timing of post-fertilization sperm mitophagy in porcine zygotes is favorable as it occurs prior to the first embryo cleavage *Sutovsky et al., 2003*; thus, this system can be utilized to study post-fertilization sperm mitophagy in a shorter time span and without interfering with the established

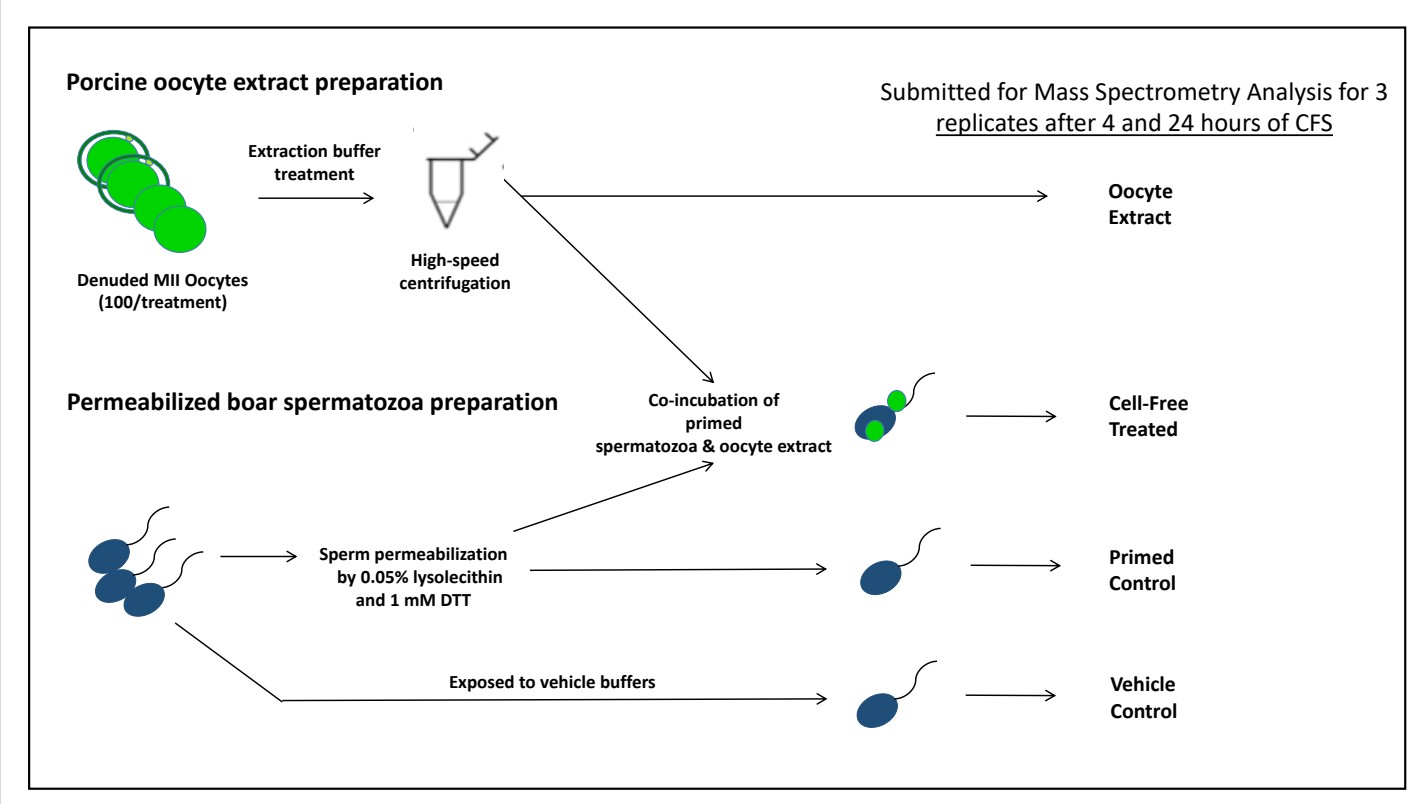

**Figure 1.** The porcine cell-free system and workflow diagram for the preparation of samples for mass spectrometry analysis.

role of the ubiquitin-proteasome system in cell division, a feat that would be more complicated in a bovine embryo where the post-fertilization sperm mitophagy occurs between the two- and four-cell stage (*Sutovsky et al., 1999*). Furthermore, *X. laevis* egg extracts do not lend themselves to the study of mammalian sperm mitophagy as it is a species-specific recognition and degradation process. To prepare oocyte extracts for this system, MII oocytes are denuded of their cumulus cells and *zona pellucida*, then placed in an extraction buffer and subjected to three rounds of flash freezing followed by thawing to disrupt their cellular membranes. At that point, the oocytes are centrifuged at high speed to be crushed, and the supernatant present is collected. This oocyte extract contains cytoplasmic proteins and some of the lysosomal and autophagosomal membrane fractions that participate in mitophagy. The paternal component of this cell-free system is represented by boar spermatozoa primed through a demembranating treatment with lysophosphatidylcholine (lysolecithin) followed by disulfide bond reduction via dithiothreitol (DTT), a stepwise treatment which removes the plasma and outer acrosomal membranes of the spermatozoa and destabilizes the structural sperm proteins in a fashion similar to sperm demembranation during sperm-oocyte fusion and disulfide bond reduction in the sperm head and tail at the time of sperm incorporation in the oocyte cytoplasm. Thus, this chemical demembranation and destabilization is used to replicate in vivo spermatozoa processing during natural fertilization. These primed spermatozoa are then co-incubated with the oocyte extract for 4–24 hr and early fertilization-specific proteomic interactions can be recapitulated. The porcine cell-free system allows us to observe thousands of spermatozoa interacting with ooplasmic proteins in a single trial, thus overcoming the limiting factor of one spermatozoon per fertilized egg, as seen in in vitro fertilization (IVF) and intracytoplasmic sperm injection (ICSI) protocols. This cell-free system has been previously shown to recapitulate fertilization sperm mitophagy events which take place in a zygote (*Song and Sutovsky, 2018*; *Song et al., 2021*).

In this study, the porcine cell-free system was used in conjunction with MADLI-TOF mass spectrometry to conduct a quantitative investigation of early fertilization proteomics (*Figure 1*). Two different trials were conducted each containing biological triplicates and based on this data an inventory of 185 proteins (p<0.1) of potential interest in the context of post-fertilization sperm mitophagy

was compiled. Six of these proteins were further investigated, including major vault protein (MVP), proteasomal assembly chaperone 2 (PSMG2), proteasomal subunit alpha 3 (PSMA3), FUN14 domain-containing protein 2 (FUNDC2), sorting and assembly machinery component 50 (SAMM50), and BAG family molecular chaperone regulator 5 (BAG5). These six proteins were considered candidate mitophagy proteins of interest based on their known functions in established autophagy-related pathways. The investigation was an attempt to understand these proteins in more detail than what was extrapolated from the mass spectrometry data. We once again used the porcine cell-free system, but this time in conjunction with immunocytochemistry (ICC) and western blotting (WB), to characterize the localization and modification changes these proteins underwent within the system. Furthermore, we investigated these proteins in zygotes after IVF to characterize their localization patterns during in vitro fertilization.

## Results

### Quantitative proteomics with the cell-free system

MALDI-TOF mass spectrometry was used to analyze spermatozoa which were exposed to the porcine cell-free system. The goal of this mass spectrometry trial was to capture changes in protein quantities between control, primed spermatozoa samples (no extract exposure), and cell-free system treated sperm samples. Vehicle control sperm samples were also submitted, as were samples of oocyte extract. A workflow diagram of the study is shown in *Figure 1*. Three biological replicates of this trial were submitted to compare primed control vs cell-free treated sperm after 4 hr of cell-free system co-incubation; separately, three biological replicates were submitted to compare primed control vs cell-free treated spermatozoa after 24 hr of cell-free system co-incubation. During both trials, three biological replicates of vehicle control sperm and oocyte extract were submitted as well. Raw data captured from mass spectrometry (*Supplementary file 1*) was referenced against the *Sus scrofa* UniProt Knowledge base, thus capturing an inventory of proteins present in each sample and their relative abundance. The samples were normalized based on the content of outer dense fiber protein 1, 2, and 3 and then subjected to statistical analysis. The primed control and cell-free-treated sperm samples were statistically compared by using a paired T-test spectrometry (*Supplementary file 1*). This T-test compared the relative normalized protein abundance between the primed control and cell-free-treated samples. A p-value of 0.2 (class 1), or 0.1 (class 2 and 3) was considered statistically relevant for the purpose of this study.

After T-test analysis in the 4-hr trial, 138 proteins were found to undergo changes (p<0.1) in abundance between the primed control vs. cell-free-treated spermatozoa. In the 24 hour trial, 56 proteins were (p<0.1) different in abundance between the primed control and cell-free-treated spermatozoa. Of these significant proteins from each trial, 14 overlapped, resulting in a total of 180 statistically different (p<0.1) proteins identified between the two trials. Between the two trials, 24 proteins were only found in cell-free-treated sperm samples and were not present in the primed control samples. These proteins were assumed to be proteins from the oocyte extract which remained bound to spermatozoa after extract co-incubation. For proteins that followed this pattern, we loosened our statistical parameters and included proteins in our inventory out to p<0.2. In the 4-hr trial, this resulted in 6 more proteins being included, and in the 24-hr trial, this resulted in seven additional proteins. Of these 13 proteins, 8 overlapped; thus, this inclusion step added 5 more proteins to our inventory for a grand total of 185 proteins. The 4 hr inventory ultimately included 144 proteins of interest (*Supplementary file 2*), whereas the 24 hr inventory contained 63 proteins of interest (*Supplementary file 3*). These inventories had an overlap of 22 proteins. The full data sheets including all reps for the 4- and 24 hr trials as well as the normalizations and T-test analyses can be found in *Supplementary file 1*.

Both the 4 hr and 24 hr protein inventories were divided into three different classes. Class 1 proteins were detected only in the oocyte extract (absent in the vehicle control and primed control spermatozoa) and found on the spermatozoa only after extract co-incubation. These proteins are interpreted as ooplasmic mitophagy receptors/determinants and nuclear/centrosomal remodeling factors. Class 2 proteins were detected in the primed control spermatozoa but increased in the spermatozoa exposed to cell-free system co-incubation. Class 3 proteins were present in both the gametes or only the primed control spermatozoa, but are decreased in the spermatozoa after co-incubation, interpreted as sperm-borne proteolytic substrates of the oocyte autophagic system. These complete protein

inventories, sorted by class for both the 4- and 24 hr trials, can be found in *Supplementary files 2 and 3*, respectively.

The functions of all the proteins added to these inventories were manually searched and categorized by using the UniProt Knowledgebase as well as PubMed literature search. Based on known functions either in the gametes or somatic cells, all proteins were categorized, and pie charts were rendered (*Figure 2A–F*; *Figure 2—figure supplements 1–6*). It should be noted that our Mass Spectrometry generated data was analyzed for proteins by using the UniProt Knowledgebase and was only analyzed for known *Sus scrofa* proteins.

## Investigation of candidate proteins in the porcine cell-free system

Six candidate proteins were selected from the mass spectrometry results for further investigation. These six proteins were MVP, PSMG2, PSMA3, FUNDC2, SAMM50, and BAG5. Western blot detection and immunocytochemistry were used to describe the presence and localization patterns of these candidate proteins in ejaculated (non-capacitated), primed, and cell-free-treated spermatozoa. Furthermore, immunocytochemistry was used to observe the localization of these proteins in in vitro derived porcine zygotes. Oocytes were fertilized with spermatozoa pre-labeled with MitoTracker so that mitochondrial sheaths could be detected. These oocytes, now presumed zygotes, were then collected at 15 and 25 hr post insemination (sperm and oocyte mixing). These time points were selected to ensure that post-fertilization sperm mitophagy was well underway in the zygotes at both timepoints. Furthermore, the 25 hr time point was selected to observe the advanced stages of mitophagy post-gamete mixing. These presumed zygotes were then fixed and stained for immunocytochemistry.

### MVP

Major vault protein was identified as a Class 1 protein of interest in our mass spectrometry trials. It was not identified in vehicle control spermatozoa or primed control sperm samples. However, it was identified in oocyte extracts and in spermatozoa exposed to the cell-free system at both the 4- and 24 hr times points. Upon further investigation, we confirmed that MVP was not detected in ejaculated spermatozoa by using both Western blotting and immunocytochemistry detection methods (*Figure 3A and B*). Additionally, after priming the spermatozoa, MVP was still not detected (*Figure 3C*). However, consistent with mass spectrometry observations, MVP was detected on the tails of treated spermatozoa after both 4 and 24 hr of cell-free system exposure (*Figure 3D and E*). In zygotes 15 hr post insemination, MVP was detected abundantly in the cytoplasm as reported previously (*Sutovsky et al., 2005*) though it did not appear to be associating with forming male pronuclei (PN) or with the mitochondrial sheaths of the fertilizing spermatozoa (*Figure 3F*). At 25 hr post insemination, MVP was still detectable in the cytoplasm of the zygote and did appear to have colocalization on the mitochondrial sheath of the fertilizing spermatozoa (*Figure 3G*).

### PSMG2

During our mass spectrometry trials, proteasome assembly chaperone 2 was identified as a Class 2 protein during the 24-hr trial. PSMG2 was found to undergo a significant increase in abundance after 24 hr in the cell-free system (p=0.088). However, during the 4-hr trial, PSMG2 increased during two of the replicates but underwent a decrease in one of the replicates and was thus not found to be significant. PSMG2 was detected in ejaculated spermatozoa via WB (*Figure 4A*) and found to localize to the acrosome of ejaculated spermatozoa (*Figure 4B*). After priming, PSMG2 was still detected in the head of spermatozoa (*Figure 4C*). After 4 hr of cell-free system co-incubation PSMG2 was detected in both the head and principal piece of the sperm tail (*Figure 4D*). After 24 hr of cell-free system exposure, this tail localization was no longer present and PSMG2 is found only on the partially decondensed (as a result of oocyte extract inducing sperm nucleus remodeling) heads of the spermatozoa (*Figure 4E*). When observed in zygotes 15 hr post insemination PSMG2 had begun to cluster around the newly forming paternal, sperm-derived pronuclei (*Figure 4F*). At 25 hr post insemination, a robust cluster of PSMG2 was detected surrounding the male pronuclei and on the mitochondrial sheath of the fertilizing spermatozoa (*Figure 4G*).

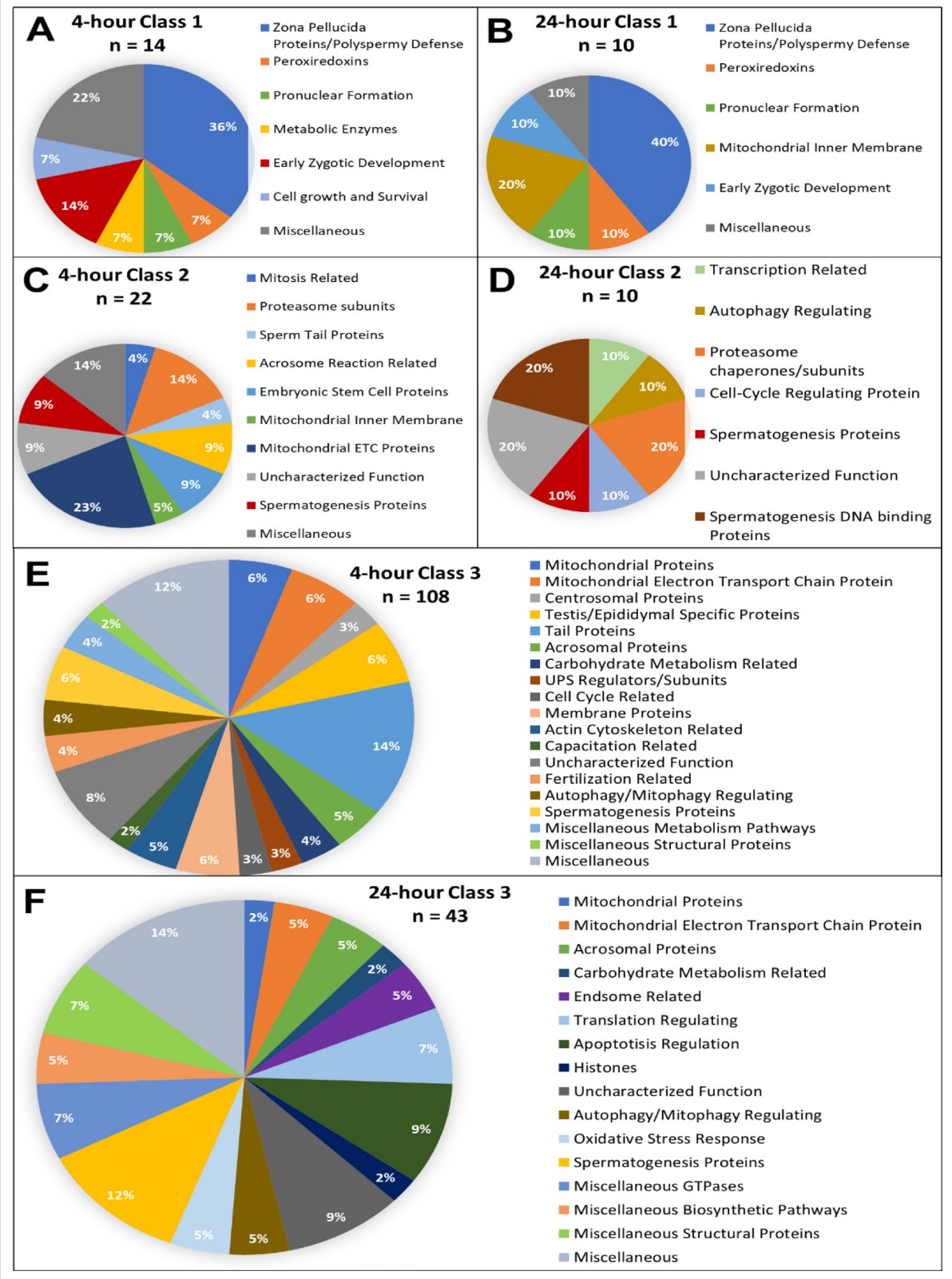

**Figure 2.** Candidate protein categorization by their functions. Proteins found in different amounts after cell-free system coincubation at both 4 and 24 hr are categorized by function as found on Uniprot.org and literature via PubMed. Protein characterizations of Class 1 found only in cell-free treated spermatozoa samples after 4 hr of cell-free system co-incubation (**A**), and 24 hr of cell-free system co-incubation (**B**) vs primed control spermatozoa (p<0.2). Characterization of Class 2 proteins which underwent an increase in abundance (p<0.1) in cell-free-treated spermatozoa during the 4 hr (**C**) and

*Figure 2 continued on next page*

*Figure 2 continued*

24 hr (**D**) cell-free system trials vs primed control spermatozoa. Class 3 proteins, which underwent a decrease in abundance (p<0.1) in cell-free-treated spermatozoa after 4 hr (**E**) and 24 hr (**F**) of cell-free system co-incubation vs primed control spermatozoa.

The online version of this article includes the following figure supplement(s) for figure 2:

**Figure supplement 1.** Proteins found in different amounts after Cell-free coincubation at both 4 and 24 hr categorized by function as found on Uniprot.org and literature via Pubmed.com.

**Figure supplement 2.** Proteins found in different amounts after Cell-free coincubation at both 4 and 24 hr categorized by function as found on Uniprot.org and literature via Pubmed.com.

**Figure supplement 3.** Proteins found in different amounts after Cell-free coincubation at both 4 and 24 hr categorized by function as found on Uniprot.org and literature via Pubmed.com.

**Figure supplement 4.** Proteins found in different amounts after Cell-free coincubation at both 4 and 24 hr categorized by function as found on Uniprot.org and literature via Pubmed.com.

**Figure supplement 5.** Proteins found in different amounts after Cell-free coincubation at both 4 and 24 hr categorized by function as found on Uniprot.org and literature via Pubmed.com.

**Figure supplement 6.** Proteins found in different amounts after Cell-free coincubation at both 4 and 24 hr categorized by function as found on Uniprot.org and literature via Pubmed.com.

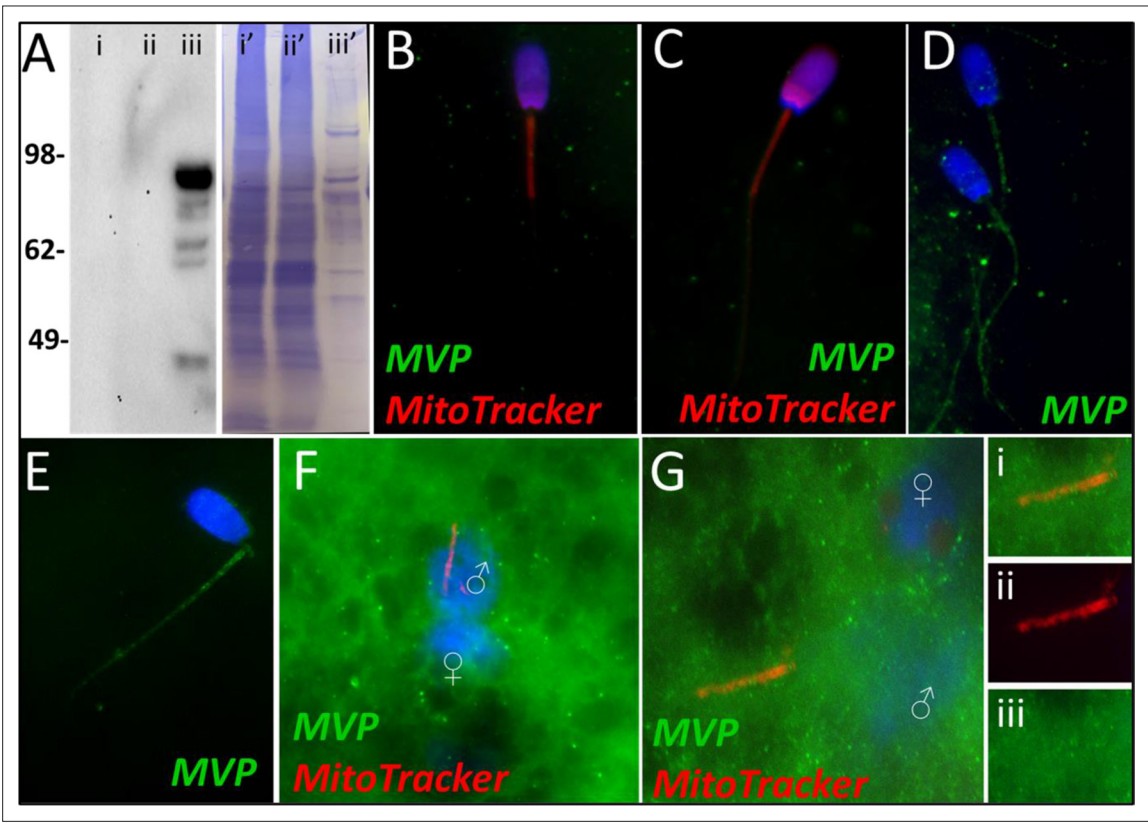

**Figure 3.** MVP in the porcine cell-free system. Major vault protein was not detected in ejaculated (Lane i) or capacitated spermatozoa (Lane ii) via Western blot detection, but it was detected in oocyte extract (Lane iii) (**A**; predicted mass = 99 kDa); (Lanes i', ii', iii') On the right side of panel A, PVDF membrane stained with Coomassie brilliant blue after chemiluminescence detection shows protein loads within each lane. MVP was not detected in ejaculated (**B**) or primed spermatozoa (**C**) via immunocytochemistry. After both 4 and 24 hr of cell-free system exposure, MVP (green) was then detected throughout the tail of the treated spermatozoa (**D and E**). MVP was detected in the cytoplasm of zygotes 15 hr post insemination, but did not appear to associate with the pronuclei or mitochondrial sheath (**F**). MVP was still detected in the cytoplasm of zygotes 25 hr post insemination, and did appear to have some association with the mitochondrial sheath of the fertilizing spermatozoa (**G**). A zoomed-in cutout of the mitochondrial sheath can be found in (**Gi**). The mitochondrial sheath is shown by MitoTracker labeling in the red channel separation panel (**Gii**). The green/protein labeling channel separation is shown in (**Giii**).

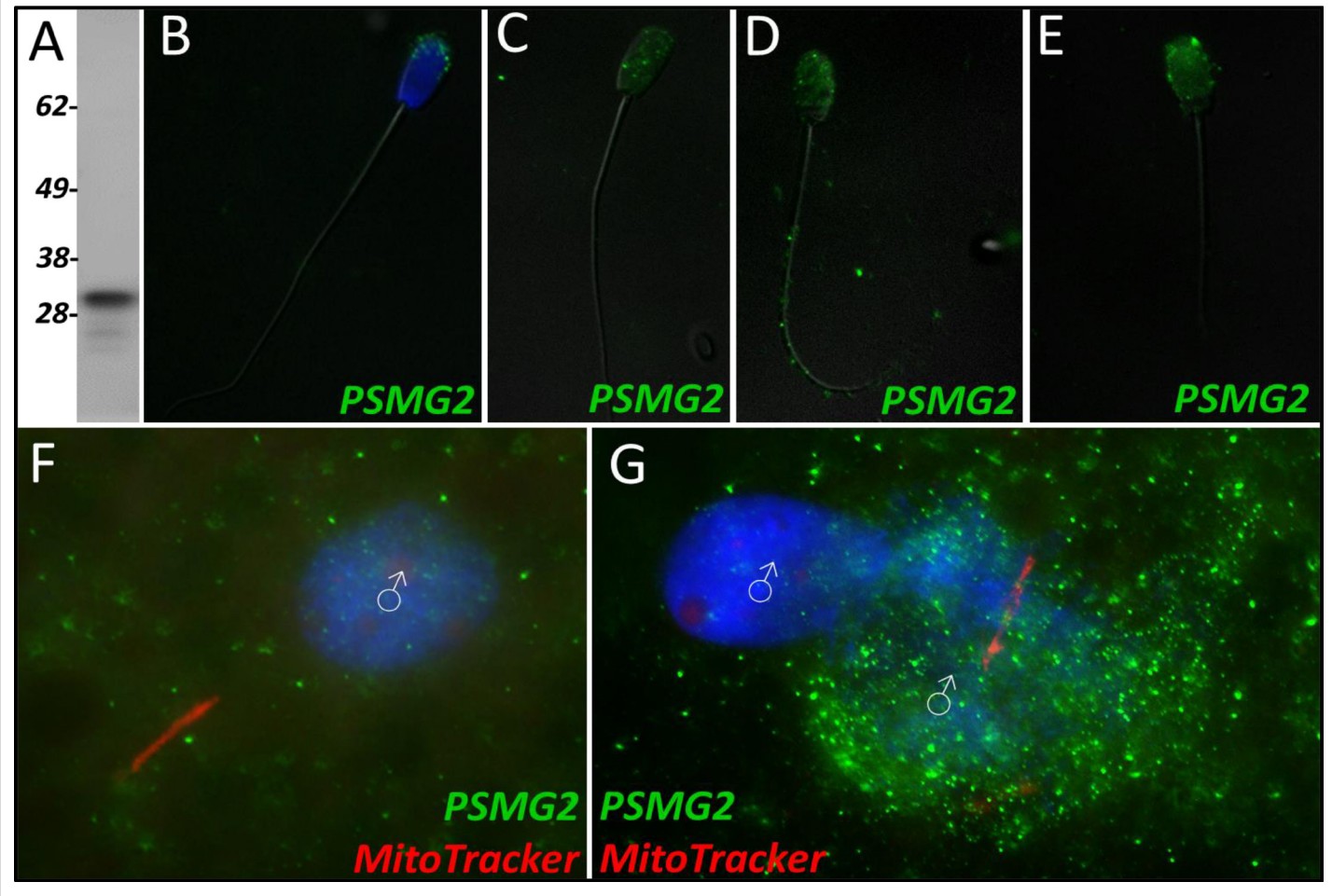

**Figure 4.** PSMG2 in the porcine cell-free system. Proteasomal assembly chaperone 2 was detected in ejaculated spermatozoa using Western blotting (**A**; predicted mass = 29 kDa), and immunocytochemistry (green) (**B**), where it was found to localize to the acrosome. In primed spermatozoa, PSMG2 was spread throughout the entire head (**C**). After 4 hr of cell-free system exposure, this same localization pattern was detected in the sperm head, but PSMG2 was also localized to the principal piece of the tail (**D**). After 24 hr of cell-free system exposure, PSMG2 was only detected in the head of the treated spermatozoa (**E**). PSMG2 was detected around the newly forming paternal pronuclei in zygotes 15 hr post insemination (**F**). PSMG2 is detected robustly localizing to the paternal pronuclei and the mitochondrial sheath of the fertilizing spermatozoa in this polyspermic zygote, 25 hr post insemination (**G**). Note that the larger, more developed PN on the right has the majority of the PSMG2 labeling.

## PSMA3

Proteasome subunit alpha 3 significantly increased (p=0.015) during the 4 hr mass spectrometry trials and was categorized as a Class 2 protein. During our 24 hr mass spectrometry trial, PSMA3 was found to undergo an increase during two of the replicates but displayed a decrease in one of the replicates. PSMA3 was detected in ejaculated spermatozoa by using both WB (*Figure 5A*) and immunocytochemistry detection (*Figure 5B*); it was found to localize to the acrosome of the ejaculated spermatozoa. After priming, PSMA3 was found in the tail of the spermatozoa, including the midpiece and principal piece (*Figure 5C*). This localization pattern persisted after 4 hr of cell-free system exposure (*Figure 5D*). After 24 hr of cell-free system exposure, PSMA3 was still detected throughout the tail but with a more focused/prevalent localization pattern on the mitochondrial sheath (*Figure 5E*). In zygotes 15 hr post insemination, PSMA3 clustered around the nascent paternal pronuclei and on the mitochondrial sheath of the fertilizing spermatozoa (*Figure 5F*). This same localization pattern persisted 25 hr post insemination and PSMA3 continued to cluster near the male and female pronuclei and on the mitochondrial sheath of fertilizing spermatozoa (*Figure 5G*).

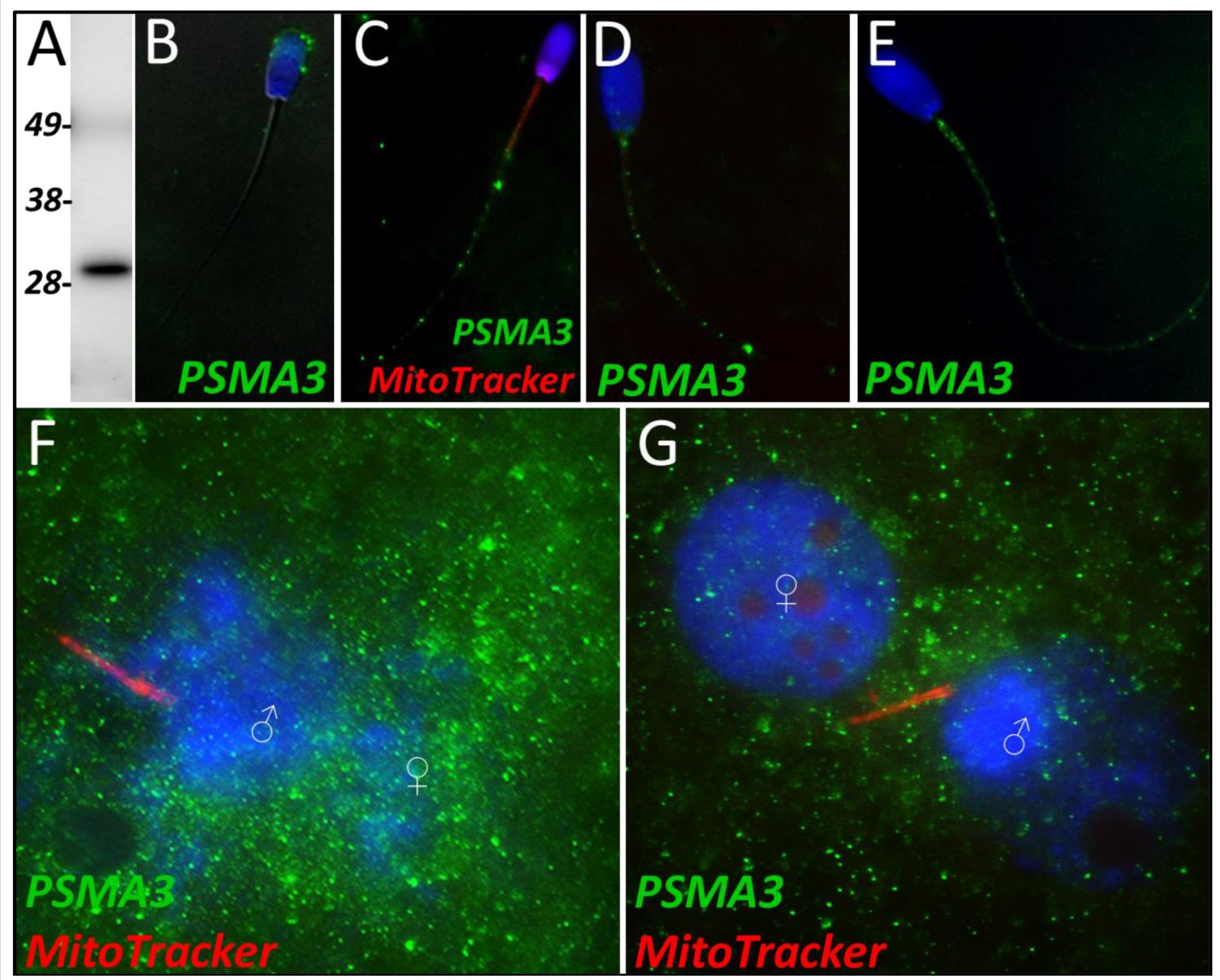

**Figure 5.** PSMA3 in the porcine cell-free system. Proteasomal subunit alpha 3 was detected in ejaculated spermatozoa via Western blotting detection (**A**; predicted mass = 28.4 kDa). PSMA3 (green) was localized to the acrosome of ejaculated spermatozoa by immunocytochemistry (**B**). After the priming process, PSMA3 was detected in the tail of primed spermatozoa (**C**). This same tail localization pattern was observed after both 4 and 24 hr of cell-free system exposure (**D and E**). PSMA3 was detected in zygotes both 15 and 25 hr post insemination (**F and G**); it was detected surrounding the male and female pronuclei as well as on the mitochondrial sheath of the fertilizing spermatozoa.

## FUNDC2

FUN14 domain-containing protein 2, a Class 3 protein in our classification, was observed to undergo a significant decrease in abundance after 4 hr of cell-free system exposure (*P*=0.084). No significant change in the abundance of FUNDC2 was detected during the 24-hr trial. During our 24-hr mass spectrometry trial, FUNDC2 was found to undergo a reduction during two of the replicates but displayed a slight increase in one of the replicates. FUNDC2 was detected in ejaculated spermatozoa via WB detection (*Figure 6A*). In ejaculated spermatozoa, FUNDC2 was detected in the acrosome and equatorial segment of the sperm head and was confined to the mitochondrial sheath within the sperm tail (*Figure 6B*). After sperm capacitation, FUNDC2 changed its localization and was found primarily in the apical ridge of the acrosome but also in the mitochondrial sheath (*Figure 6C*). After the removal of disulfide bonds via priming, FUNDC2 was predominantly detected in the mitochondrial sheath of primed spermatozoa compared to ejaculated and capacitated spermatozoa and was still detected

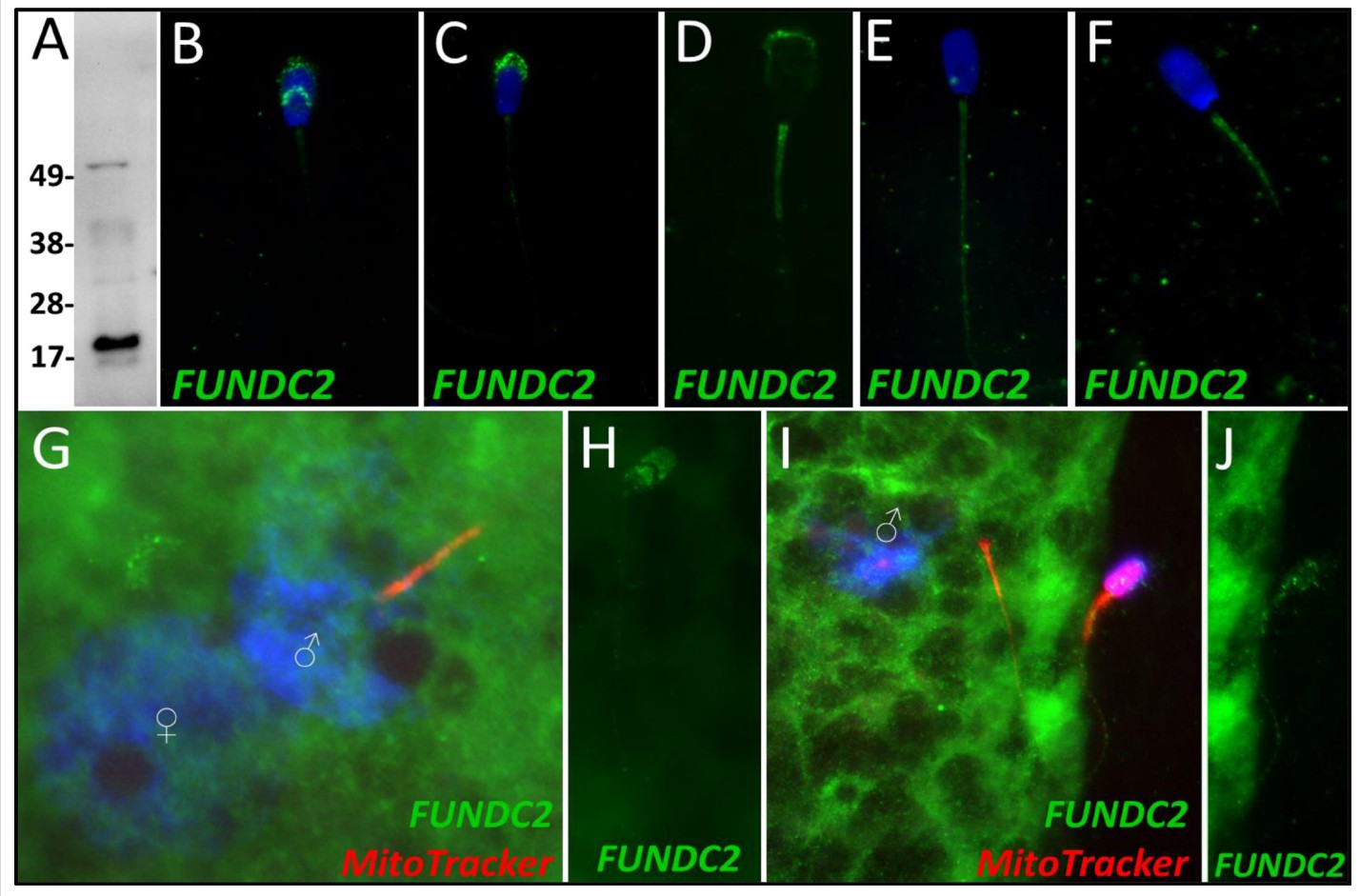

**Figure 6.** FUNDC2 in the porcine cell-free system. FUN14 domain-containing protein 2 was detected in ejaculated spermatozoa by Western blotting (**A**; predicted mass = 20.7 kDa). FUNDC2 was detected in the acrosome and equatorial segment of ejaculated spermatozoa (**B**) and the acrosome and mitochondrial sheath of capacitated spermatozoa (**C**). In primed spermatozoa, FUNDC2 was detected in the mitochondrial sheath and the remnants of the acrosome (**D**). After 4 hr of cell-free system exposure, FUNDC2 was detected throughout the tail of the treated spermatozoa (**E**), and after 24 hr of cell-free system exposure, FUNDC2 was detected in the mitochondrial sheath of treated spermatozoa (**F**). FUNDC2 was not detected localizing to the fertilizing sperm components in zygotes 15 hr post insemination (**G**). In contrast, a zona-bound spermatozoon at that same time point had FUNDC2 localized throughout the head and some tail localization as well (**H**). FUNDC2 was not detected localizing to the fertilizing sperm components in zygotes 25 hr post insemination (**I**), but a non-fertilizing spermatozoon bound to the oolemma at this same time point still had some FUNDC2 labeling in its tail and head, as shown by a fluorescent channel separated cutout (**J**).

on the remnants of the acrosome (***Figure 6D***). Upon 4 hr of cell-free system exposure, FUNDC2 was found to localize throughout the entire tail of the spermatozoa (***Figure 6E***). Interestingly, after 24 hr of cell-free system exposure, FUNDC2 was again confined to the mitochondrial sheath of the spermatozoa but appeared to have a fractured pattern, perhaps revealing the remaining mostly intact mitochondria or vice versa, that is, it may be localized to those mitochondria which have undergone the greatest degree degradation at this time point (***Figure 6F***).

FUNDC2 was not detected on or near the fertilizing spermatozoa at 15 hr post insemination (***Figure 6G***); in contrast, FUNDC2 was still detected in the head and tail of a spermatozoon bound to the zona pellucida of a zygote at the same time point (***Figure 6H***). FUNDC2 was also not observed localizing on or near the fertilizing spermatozoa at 25 hr post insemination, but was detected in non-fertilizing, zona-bound spermatozoa at the same time point (***Figure 6I and J***).

## SAMM50

Sorting and assembly machinery component 50 was observed to undergo a significant decrease in abundance during the 4 hr mass spectrometry trial (p=0.035), classifying it as a Class 3 protein. This

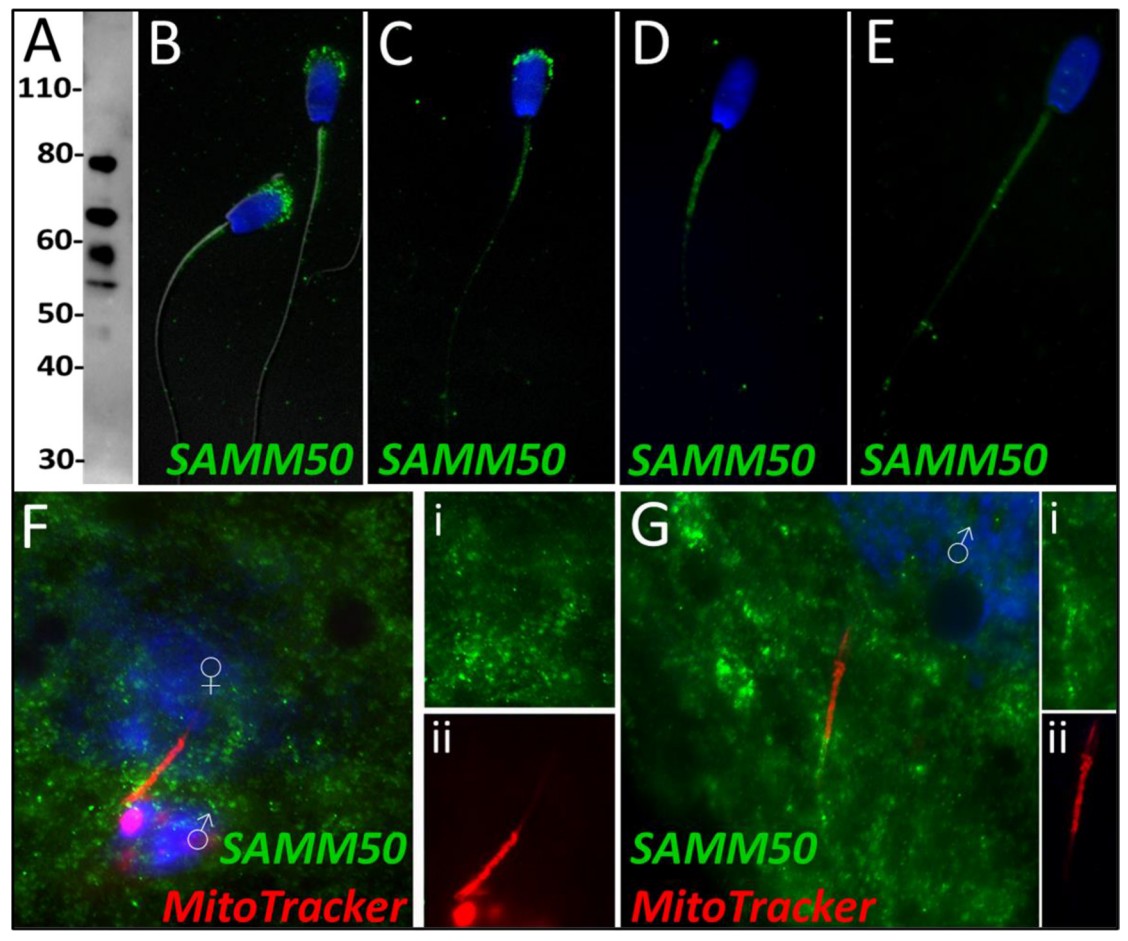

**Figure 7.** SAMM50 in the porcine cell-free system. Sorting and assembly machinery component 50 was detected in ejaculated spermatozoa using Western blotting (**A**; predicted mass = 51 kDa). SAMM50 has predicted post translational modifications, including phosphorylation sites and ubiquitination sites, as predicted using the MuSite Deep prediction software which likely explains the higher bands which are observed. SAMM50 was also detected in ejaculated spermatozoa using immunocytochemistry and found to localize to the acrosome and the mitochondrial sheath of the sperm tail (**B**). In primed spermatozoa, SAMM50 was detected in the remnants of the acrosome, as well as in the mitochondrial sheath, and the principal piece (**C**). After 4 hr of cell-free system exposure, SAMM50 was detected predominantly in the mitochondrial sheath with some signal present in the principal piece of the sperm tail as well (**D**). After 24 hr of cell-free system exposure, SAMM50 was detected predominantly in the mitochondrial sheath, still with residual principal piece labeling as well (**E**). SAMM50 in zygotes 15 hr post insemination was detected on and near the mitochondrial sheath of the fertilizing spermatozoa (**F**). Fluorescence channel separation cutouts of the mitochondrial sheath and principal piece are shown in (**Fi**; green SAMM50) and (**Fii**; red MitoTracker). SAMM50 was also detected in zygotes 25 hr post insemination on the principal piece of the tail of the fertilizing spermatozoa, just below the mitochondrial sheath (**G**). Fluorescence channel separation cutouts of the mitochondrial sheath and principal piece remnants are shown in (**Gi**; green SAMM50) and (**Gii**; red MitoTracker).

decrease in abundance was not observed during the 24 hr mass spectrometry trial wherein SAMM50 underwent a reduction within two of the replicates but underwent a slight increase in one of the replicates. SAMM50 was detected in spermatozoa via WB and immunocytochemistry (**Figure 7A and B**); it was found to localize to the acrosome and mitochondrial sheath of the sperm tail. After priming, this same localization pattern persisted on the mitochondrial sheath and in what remained of the acrosome and appeared to localize to the principal piece as well (**Figure 7C**). After 4 hr of co-incubation within the cell-free system, SAMM50 was completely removed from the remnants of the acrosome, but still found throughout the tail with the greatest density of localization on the mitochondrial sheath. A similar patter was seen in primed spermatozoa as well, likely due to acrosome removal via the priming process (**Figure 7D**). This same localization pattern was observed after 24 hr of cell-free system co-incubation (**Figure 7E**). During our zygote trial, SAMM50 was observed localizing on and near the mitochondrial sheath of the fertilizing spermatozoa at 15 hr post insemination (**Figure 7F**).

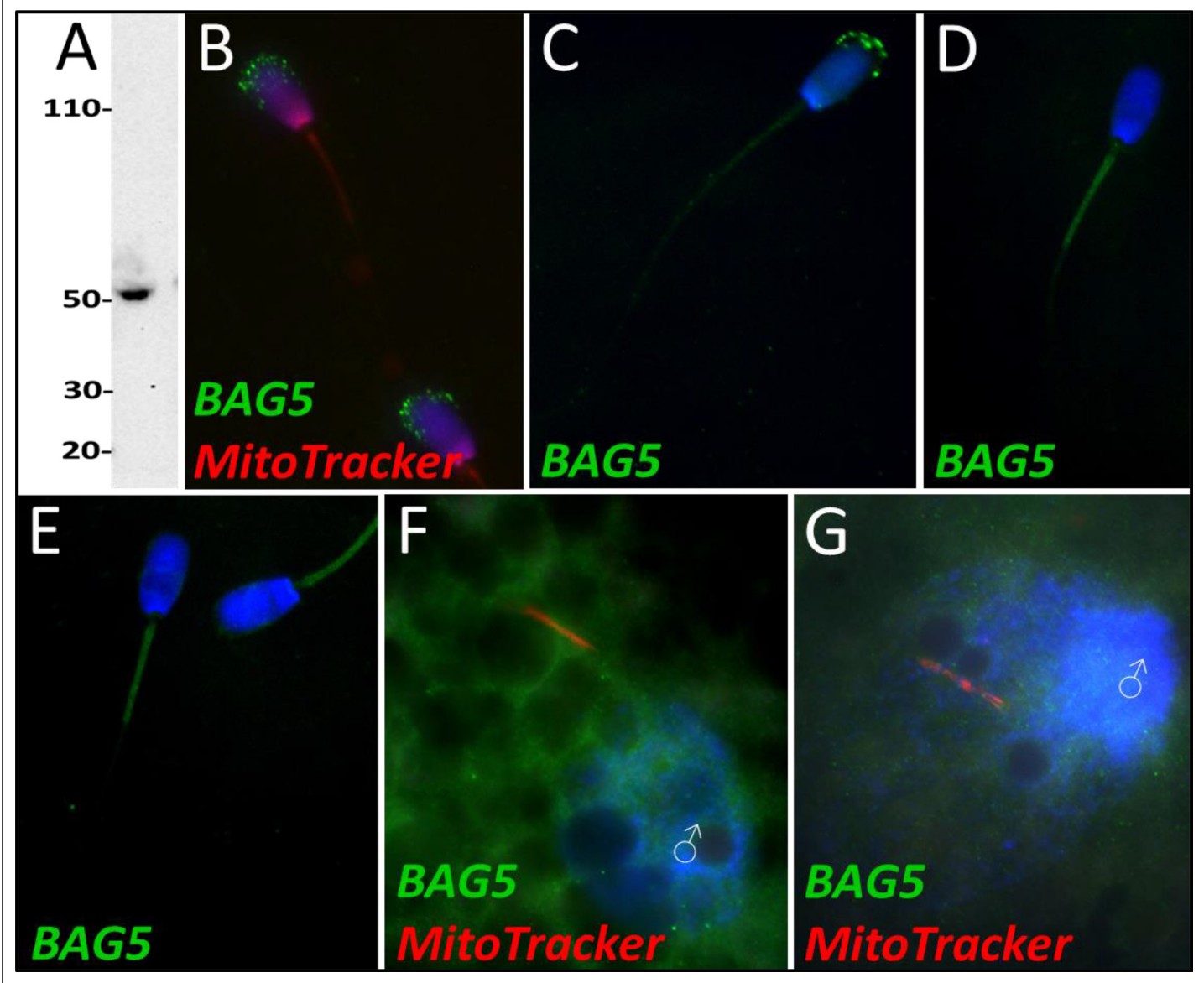

**Figure 8.** BAG family molecular chaperone regulator 5 was detected in ejaculated spermatozoa via WB (**A**; predicted mass = 54 kDa) and immunocytochemistry (**B**), where it was detected to localize in the acrosome. After priming, BAG5 still remained localized to what remained of the acrosome (**C**). After 4 and 24 hr of cell-free system exposure, BAG5 was detected in the mitochondrial sheath of the treated spermatozoa (**D and E**). BAG5 was detected in the cytoplasm in zygotes 15 hr post structures (**F**) and 25 hr post fertilization, with no obvious association with the fertilizing sperm structures or their remnants (**G**).

SAMM50 was also detected 25 hr post insemination localizing to the principal piece of the fertilizing spermatozoa, but no longer on the mitochondrial sheath (**Figure 7G**).

## BAG5

Based on quantitative proteomics, BAG family molecular chaperone regulator 5 underwent a decrease in abundance during the 4 hr mass spectrometry trial (p=0.02). However, this level of significance was not observed during the 24 hr mass spectrometry trial. During our 24 hr mass spectrometry trial, BAG5 underwent a reduction during two of the replicates but a slight increase in one of the replicates. Upon further investigation, BAG5 was detected by western blotting (**Figure 8A**) and immunocytochemistry (**Figure 8B**) in fresh ejaculated spermatozoa, where it was localized exclusively to the acrosome. After priming, BAG5 was only detectable in the remains of the acrosome and weakly on the sperm tail (**Figure 8C**). After 4 hr of co-incubation within the cell-free system, BAG5 was no longer detected on

the head of the spermatozoa but became detectable on the mitochondrial sheath of the spermatozoa (*Figure 8D*). Likewise, after 24 hr of co-incubation within the cell-free system, this same BAG5 localization to the mitochondrial sheath remained (*Figure 8E*). BAG5 was not observed localizing on the fertilizing spermatozoa 15 hr post insemination, though BAG5 did appear to be present in the cytoplasm of the zygote (*Figure 8F*). In zygotes at 25 hr post insemination, BAG5 was still not observed on the fertilizing spermatozoon but remained present within the cytoplasm of the zygote (*Figure 8G*).

## Discussion

In the last decade, considerable effort has been put forth to better understand the autophagic pathway's involvement in mitochondrial inheritance (*Al Rawi et al., 2011*; *Sato and Sato, 2011*; *Zhou et al., 2011*; *Politi et al., 2014*; *Song et al., 2016*). Much headway has been made; however, the knowledge gaps surrounding this post-fertilization sperm mitophagy process remain wide. For example, at present, it is not understood what determines species specificity or timing of sperm mitophagy. It has been observed that mammalian interspecies crosses retain the paternal mitochondria in F1 generation and become heteroplasmic as a result (*Sutovsky et al., 1999*; *Shitara et al., 1998*; *Kaneda et al., 1995*). The timing of mitochondrial degradation in embryos also varies between species (Reviewed in *Zuidema and Sutovsky, 2019*). Additionally, many protein cofactors, substrates, and autophagic pathways which act within the post-fertilization mitophagic process remain to be identified. This area of research has relied on studies with mammalian oocytes and zygotes. This led to the development of our species-specific mammalian cell-free system (*Song and Sutovsky, 2018*). This system was developed to provide a new tool for the study of post-fertilization sperm mitophagy and other early fertilization events. Specifically, the use of this system in conjunction with quantitative mass spectrometry was a type of study which has never been attempted before, even though it is becoming increasingly popular to investigate sperm proteomes by using quantitative proteomic analyses (*Baker et al., 2007*; *Martínez-Heredia et al., 2006*; *Pilatz et al., 2014*; *Zhang et al., 2022*). Attempting to identify ooplasmic proteins which bind to spermatozoa at fertilization and those sperm proteins which begin degrading during fertilization by using mass spectrometry would be exceedingly difficult by using zygotes. This is because post-fertilization sperm mitophagy takes place within the oocyte cytoplasm and the ability to identify the differences in those proteins which are found in the sperm or binding the sperm surface from the rest of the oocyte proteome would be extremely challenging/impossible since we are not yet able to reliably map and quantify single-cell proteomes. However, our cell-free system presents a unique tool which can be used to capture a number of the early proteomic changes which take place specifically on the fertilizing spermatozoa; this cell-free system has been previously shown to mimic some early fertilization events (*Song and Sutovsky, 2018*; *Song et al., 2021*). Thus, we could expect other proteomic events of early fertilization to be mimicked within our cell-free system. By using high-resolution mass spectrometry, we captured differences in relative protein abundance between primed control spermatozoa and cell-free treated spermatozoa. Furthermore, we captured this data at two different time points, that is after 4 hr and 24 hr of co-incubation within the cell-free system. Furthermore, we identified proteins of interest which can be further explored by using more targeted sperm phenotype studies.

This study was constrained to two time points, which allowed for some temporal differences to be captured. However, the differences over time in this cell-free system could be immense and the differences in the protein inventories compiled for each time point may be indicative of this. Furthermore, this cell-free system while useful does not perfectly capture all the events which take place during in vivo fertilization. The cell-free system is intended to mimic early fertilization events but is presumably not the exact same as in vitro fertilization. We recognize that both the sperm demembranating process and the oocyte extraction process resulted in a loss of excessive proteins that may also play a role in early fertilization. Also, this study captures changes in protein abundances only. Proteins, which remain at relatively stable abundances but undergo changes in localization and/or post-translational modification resulting in altered biological activity, would not be captured by the parameters set forth in this study. Despite these recognized limitations, this cell-free system used in conjunction with high resolution mass spectrometry allowed a glimpse into early fertilization proteomics in a way that has not been attempted before.

Our study ultimately identified 185 proteins which underwent a significant change in abundance within our parameters as described above. Of these proteins, 144 were identified during the 4 hr

cell-free system trial (*Supplementary file 2*) and 63 were found during the 24 hr cell-free system trial (*Supplementary file 3*), with 22 proteins overlapping between the trials. This lack of overlap was surprising but again may be reflective of our inability to truly detect the dynamic changes which take place within the cell-free system.

Considering the dynamic proteomic remodeling of both the oocyte and spermatozoa which takes place during early fertilization, these 185 proteins which have been identified likely play roles in processes beyond sperm mitophagy. Pathways related to pronuclear development, sperm aster formation, and degradation of the perinuclear theca, acrosome remnants, and tail structures including the mitochondrial sheath may all be captured. However, several autophagy-related proteins were found within our inventory including L-lactate dehydrogenase B chain, keratin 8, glycogen synthase kinase 3 beta 5, BAG family molecular chaperone regulator 5, sorting and assembly machinery component 50, and FUN14 domain-containing protein 2. Interestingly, these proteins were all found in class 3, and thus these proteins may be mitophagy regulators which must be recognized by ooplasmic mitophagy receptors and removed from spermatozoa/sperm mitochondria for mitophagy to take place. In addition to these proteins, multiple components and regulators of the ubiquitin-proteasome system were also identified including 20 S proteasomal subunits alpha type 2, 3, and 8, 26 S proteasome non-ATPase regulatory subunit 2, 26 S proteasome subunit ATPase 2, proteasome assembly chaperone 2, ubiquitin specific peptidase 50, E3 ubiquitin ligase cullin 4B and OTU deubiquitinase. These proteins are found in both classes 2 and 3; they may be involved in UPS-mediated mitophagy or play roles in the degradation of other sperm structures including the tail and perinuclear theca. There is also evidence of sperm proteasomes playing a role in acrosomal exocytosis, zona pellucida degradation, and events linked with sperm capacitation and hyperactivation (*Hackerova et al., 2023*; *Zigo et al., 2019*; *Sutovsky, 2011*; *Zimmerman et al., 2011*).

Class 1 proteins were proteins that were not identified within our primed control sperm proteome but were found in our cell-free system treated spermatozoa. We interpreted these as oocyte proteins which interacted with the oocyte extract-exposed spermatozoa and remained bound to them after thorough washing at the end of the co-incubation interval. Indeed, among the identified Class 1 proteins were several well-documented oocyte-specific proteins which are known to interact with spermatozoa during early fertilization events, before or after sperm incorporation in the oocyte cytoplasm, and are only found in oocytes. Adding to our confidence in the specificity of this cell-free proteomic system, these proteins included zona pellucida proteins 2, 3, and 4; ovastacin, DNA methyltransferase 1, and nucleoplasmin 2. Also, within Class 1, we observed proteins related to polyspermy prevention, pronuclear formation, cell development, and as well as several ROS scavenging proteins (*Figure 2A and B*). These pathways all reinforce known oocyte functions taking place during early fertilization events. Being able to capture some of these proteomic interactions is encouraging and shows that our cell-free system is indeed recapitulating specific proteomic interactions such as would be observed during in vitro fertilization.

Class 2 proteins were identified in the primed control samples but were found in an increased abundance within the cell-free system treated samples. Proteins in this class are expected to be oocyte-derived proteins which bind the sperm structures and stay bound. However, this class may also include sperm proteomic changes which are stimulated by the ooplasmic exposure but not directly attributable to oocyte proteins binding the spermatozoa. There is some evidence that during capacitation, nuclear-encoded mRNAs in the spermatozoa are translated by mitochondrial ribosomes, specifically to support capacitation, hyperactivation, acrosomal exocytosis, and fertilization. Mitochondrial proteins, sperm tail axonemal proteins, and acrosomal function-related proteins unexpectedly found in this category could be explained by this theory; this concerns cytochrome C1, cytochrome C oxidase subunit 5 A, stomatin like 2, A-kinase anchoring protein 4, NADH:ubiquinone oxidoreductase subunit B5 and B9, and Tektin 1. In fact, STOML2 is a regulator of mitochondrial translation, further providing some credence to this hypothesis. This sperm translation theory remains controversial, however, and does not necessarily explain these trends (*Gur and Breitbart, 2006*). During Class 2 protein analysis at both time points, we observed an increase in proteasomal subunits/chaperones, as discussed above. Proteins related to spermatogenesis were also identified at each time point. Within Class 2 at both time points, proteins which are known to interact with sperm nuclear DNA during spermatogenesis were identified, including testis-specific serine kinase 6, spermatogenesis associated 24, and PHD finger protein 7. These proteins could perhaps assist in hyper-condensation of the sperm nucleus

during spermatogenesis and may also play a role in the decondensation of the sperm nucleus upon fertilization. During the 4-hr trial, proteins related to the acrosome reaction, and proteins within the mitochondrial respiratory chain were also identified which is somewhat surprising as we would expect these proteins to begin to be reduced but perhaps this is an indication of some capacitation-like events still taking place early on during our cell-free system co-incubation. Proteins related to the acrosome reaction and mitochondrial respiratory chain are no longer identified at 24 hr, perhaps these proteins are early substrates of the ooplasmic protein recycling machinery (*Figure 2C and D*).

Among the Class 3 proteins, those which underwent a reduction in abundance after extract co-incubation, we observed sperm tail, acrosomal, mitochondrial, centrosome-related, spermatogenic, capacitation-related, and membrane proteins. These aforementioned structures and systems are no longer necessary after fertilization. Specifically, during the 4 hr trials, 14% of the proteins categorized as Class 3 were tail proteins, 12% were mitochondrial proteins, 6% were spermatogenesis related and 6% were testis/epididymal-specific proteins (*Figure 2E*). All these sperm protein types being degraded during early fertilization would be expected. During the 24-hr trial, 7% of proteins observed were mitochondrial, and 12% were spermatogenesis-related, supporting the early degradation of some of these substrates. Surprisingly, there were no known tail proteins identified in the 24-hr trial, although 7% of proteins were known structural proteins (*Figure 2F*). Additionally, several autophagy/mitophagy regulators were found within this class as well. Readers will note that during the 4-hr trial,

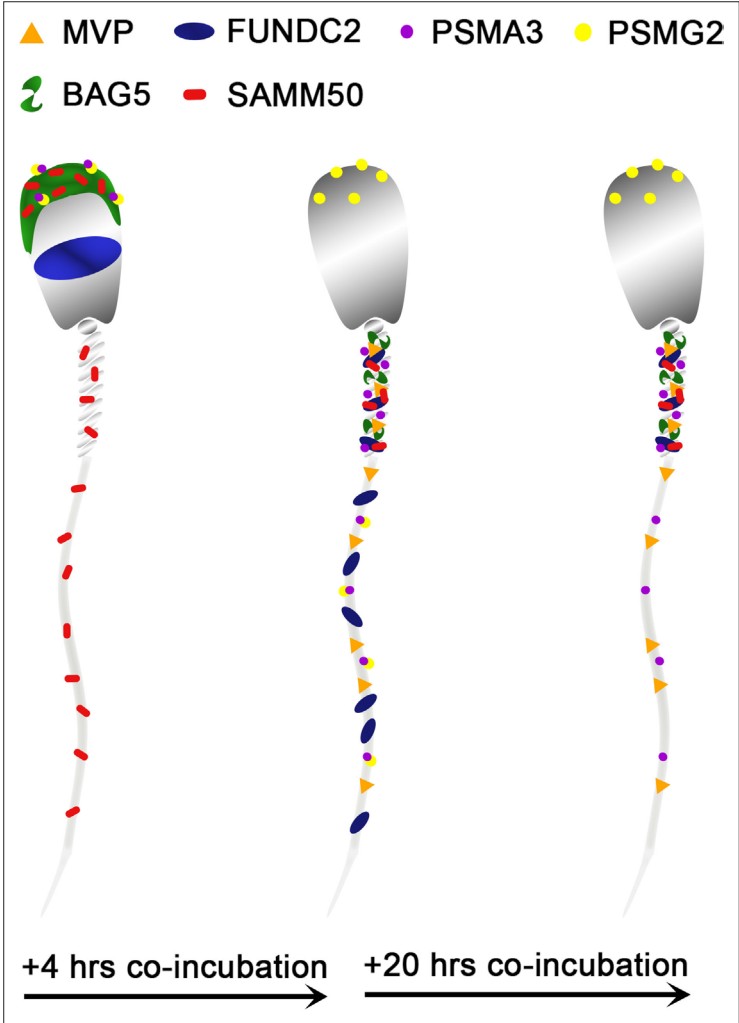

**Figure 9.** A stylized depiction of changes in protein localization patterns in boar spermatozoa over time when exposed to the porcine cell-free system. These depictions are based on the localization patterns observed in *Figures 3–8*. Shown are the observed localization patterns for MVP, FUNDC2, PSMA3, PSMG2, BAG5, and SAMM50.

108 of the 144 proteins identified, or 75%, fell into Class 3. During the 24-hr trial, 43 of the 63 identified proteins, or 68%, fell into Class 3. This means that in both trials, the majority of spermatozoa changes observed were reductions in sperm protein content. This is not surprising when considering the context of sperm structure recycling and remodeling during fertilization. Spermatozoa contribute nuclear, chromosomal DNA; a centriole, mRNAs, and a variety of proteins and other molecules to the newly forming embryo; however, the rest of the spermatozoa structures are recycled in an orderly fashion after fertilization (*Sutovsky and Schatten, 2000*; *Sutovsky, 2004*). This includes the mitochondrial sheath, as well as structural sperm tail proteins and the remnants of the acrosome, perinuclear theca, and pericentriolar compartment (*Sutovsky et al., 2004*). Spermatozoa retain many proteins which ultimately are degraded upon fertilization, a concept which appears to be further reinforced by this study.

Based on this mass spectrometry work, six candidate mitophagy proteins of interest were investigated by using the porcine cell-free system and porcine IVF in conjunction with ICC to detect protein localization patterns within each system. This was done to characterize changes in these proteins during early fertilization events, with an emphasis on exploring their potential roles in post-fertilization mitophagy. A diagram summarizing differences in the localization of these six proteins before vs after sperm esposure to the cell-free system can be found in *Figure 9*.

The first of these six protein is the major vault protein (MVP) and as its name suggests, it is the main component of the vault particle. The exact function of MVP and the vault particle remains unknown (*Sutovsky et al., 2005*; *Mossink et al., 2003*). Several functions have been speculated, including intracellular transport, ribonucleoprotein assembly, and ribonucleoprotein proteolysis (*Suprenant, 2002*). It is known, and significant to the present study that MVP turnover is regulated by the ubiquitin-proteasome system (*Sutovsky et al., 2005*). When polyubiquitinated, MVP seemed to show an affinity for recombinant UBA domain of SQSTM1/p62; however, this action was identified in the extracts of unfertilized oocytes and not zygotes (*Sutovsky et al., 2005*). The MVP degradation may be a result of increased protein turnover in zygotes or perhaps programmed destruction of maternal proteins during early fertilization (*Sutovsky et al., 2005*). Furthermore, MVP buildup is observed in low-quality human oocytes and has also been observed in abnormal porcine zygotes; thus, it appears that the degradation of both oocyte and sperm MVP during late-stage oogenesis and early fertilization may benefit proper embryogenesis, although the exact reason for this need has yet to be explained (*Sutovsky et al., 2005*). In the present study, MVP in the cell-free system followed the same pattern as detected during our mass spectrometry trial. MVP was undetectable by using Western blotting or immunocytochemistry in ejaculated and primed spermatozoa (*Figure 3A–C*). However, upon exposure to the cell-free system, MVP was detected in the tail of spermatozoa after both 4 and 24 hours of cell-free system exposure (*Figure 3D and E*), this MVP was transferred from oocyte cytosolic extract to sperm structures. This is a localization pattern we could expect from a pro-mitophagic protein, although that does not guarantee that is MVP's role here. During the zygote trials, MVP did not appear to have this same tail localization pattern at 15 hr, although in the zygotes 25 hr post fertilization (*Figure 3G*), MVP became observable on the mitochondrial sheath, though not nearly as obvious as what was seen in the cell-free system images. Besides mitophagy, the proposed roles of MVP in ribonucleoprotein assembly and degradation (*Suprenant, 2002*) are of interest after observing the localization pattern of MVP in the cell-free system, with particular emphasis on mitochondrial ribosomes. Spermatozoa are known to carry an array of RNAs that contribute to the newly formed zygote at fertilization (*Schuster et al., 2016*). This binding pattern of MVP to sperm mitochondrial sheath, observed in the cell-free system could potentially be evidence of MVP serving a role as either a ribonucleoprotein assembler or degrader, a role which may be less observable within an embryo where ample oocyte proteins are present for RNA processing. Particularly intriguing is the known affinity of polyubiquitinated MVP for autophagic ubiquitin-receptor SQSTM1, especially considering the well-documented pattern of protein ubiquitination which takes place on and around the sperm mitochondrial sheath during early fertilization (*Sutovsky et al., 1999*). MVP may be serving a role as an early binder to sperm tail structures and subsequently undergo polyubiquitination and help initiate SQSTM1 binding to these structures. These potential roles as a protein assisting in sperm RNA processing and/or assisting in signaling for pro-autophagic SQSTM1 or perhaps a dual role where it performs both tasks would be interesting as previously unknown roles for MVP in the context of zygotic development.

Proteasomal assembly chaperone 2 participates in proteasome formation. The 26 S proteasome is a large protease complex which is composed of a 20 S catalytic core particle (CP) and one or two 19 S regulatory particles or an 11 S activator complex. The 26 S proteasome is responsible for degrading polyubiquitinated proteins into small peptides. The 20 S CP is formed by two outer α-rings and two inner β-rings (αββα arrangement). The alpha ring and beta ring are each composed of 7 alpha or beta subunits, respectively. The formation of these rings and the whole CP is mediated by proteasome assembly chaperones PSMG1-4. PSMG2 is known to form a heterodimer with PSMG1; together these two proteins act as a quality control chaperone complex during CP assembly (*Wu et al., 2018*). PSMG1-PSMG2 associate with the α-ring and ensure proper CP formation, and while doing so block nuclear translocation of nascent CP, thus preserving the retention of α-ring intermediates in the cytoplasm, and only allowing mature CPs to enter the nucleus of cells (*Wu et al., 2018*). The 26 S proteasome is known to degrade mitochondrial proteins during post-fertilization sperm mitophagy; thus, identifying a protein responsible for the assembly of proteasomes undergoing an increase in abundance within our cell-free system was unsurprising. Regarding this study, PSMG2 was continuously detected on the head of the spermatozoa throughout extract coincubation (*Figure 4A–E*), though after 4 hr of cell-free system exposure it was present on the principal piece of the tail as well. This localization pattern becomes even more interesting upon observing PSMG2 in zygotes, wherein PSMG2 clusters around the paternal pronucleus at both time points (*Figure 4F and G*) and clustered around the mitochondrial sheath of the fertilizing spermatozoa in 25 hr zygotes. The 26 S proteasome has been previously described as exhibiting this same localization pattern in porcine embryos (*Huo et al., 2004*), and the 26 S proteasome is considered crucial for proper PN function. Thus, it is unsurprising that the protein responsible for the proteasome assembly is also present around the PN. Furthermore, the UPS has been implicated in post-fertilization sperm mitophagy (*Sutovsky et al., 2003*), which explains the clustering that can also be observed around the mitochondrial sheath of the fertilizing spermatozoa. It is worth noting how much more robustly PSMG2 performs during in vitro fertilization as compared to the cell-free system. Though PSMG2 appears to be positioning on the head of the spermatozoa and the tail during the 4 hr cell-free trial, the difference in zygotes is profound.

Proteasomal Subunit Alpha 3 (PSMA3) is one of the 7 subunits which comprise the 20 S CP's outer alpha rings. However, the 26 S proteasome is composed of at least 33 different subunits, organized into two particles, the 20 S core particle (CP) and the 19 S regulatory particles (*Pack et al., 2014*). Interestingly, PSMA3 is one of the three alpha subunits which incorporate into the alpha ring independently (PSMA1, PSMA2, and PSMA3), while the other four alpha subunits are incorporated by the PSMG1-PSMG2 complex. Though we did not investigate it further here, PSMA2 also underwent a significant increase in abundance during the 4 hr mass spectrometry trial. Our trial appeared to capture the majority of the chaperone-independent alpha subunits (PSMA2 and PSMA3) and half of the complex which regulates the dependent alpha subunits (PSMG2), all increasing in abundance. This confirms that the proteasome assembly pathways are active in the cell-free system. Likely, a combination of sperm-borne and ooplasm-derived proteasome subunits and regulators are observed and interacting within the cell-free system. The ubiquitin-proteasome system has already been implicated in post-fertilization sperm mitophagy (*Sutovsky et al., 1999*), as previously mentioned; thus, the increase of these subunits and regulators is to be expected. In spermatozoa, the sperm acrosome contains a plethora of 26 S proteasomes and consequently the various proteasomal subunits, including PSMA3 *Sutovsky, 2011*; this acrosomal localization pattern can be observed for both PSMG2 (*Figure 4B*) and PSMA3 in ejaculated spermatozoa (*Figure 5B*), although sperm priming for cell-free systems may remove most of the proteasomes residing in the outer acrosomal membrane and acrosomal matrix, and only leave the proteasomal subunits of the extraction-resistant inner acrosomal membrane intact (*Yi et al., 2010*). However, PSMA3 has a localization pattern distinct from PSMG2 both in the primed and in cell-free system-treated spermatozoa (*Figure 5C–E*). PSMA3 is present in the tail after priming and cell-free system exposure at both time points. However, within zygotes, in a fashion similar to PSMG2, PSMA3 has a much more robust localization pattern. PSMA3 clusters around both pronuclei and is detected on the mitochondrial sheath of the fertilizing spermatozoa (*Figure 5F and G*). As previously mentioned, the ubiquitin-proteasome system is known to assist with PN development and mitochondrial sheath degradation, so this localization pattern within zygotes is expected. Such proteasome assembly topology could go hand-in-hand with the activity of ooplasmic valosin containing protein (VCP), thought to assist with the extraction and delivery of

the ubiquitinated sperm mitochondrial membrane proteins to ooplasmic proteasomes (*Song et al.,* *2016*).

FUNDC2 is a homolog of the known mitophagic protein FUNDC1. FUNDC2 has been found to localize to the outer membrane of the mitochondria, where it serves an anti-apoptotic role through interactions with AKT1 (*Ma et al., 2019*). FUNDC2 has also been predicted but not described as playing a role in mitophagy. FUNDC2 was detected in the acrosome of ejaculated and capacitated spermatozoa but in clearly different localization patterns (*Figure 6B and C*). After disulfide bond removal, FUNDC2 becomes detectable in the mitochondrial sheath as well (*Figure 6D*). FUNDC2 was detected throughout the tail after 4 hr of cell-free system exposure (*Figure 6E*) and then once again only in the mitochondrial sheath of spermatozoa exposed to 24 hr of the cell-free system (*Figure 6F*). The loss of acrosomal membrane and matrix during cell-free system co-incubation likely explains the mass spectrometry observed drop in FUNDC2 abundance. FUNDC2 was not observed on the fertilizing spermatozoa in zygotes either at 15 or 25 hr post insemination (*Figure 6G and I*); this was observed in clear contrast to zona-bound spermatozoa at each time point, which retained FUNDC2 content (*Figure 6H and J*). The removal of FUNDC2 from sperm mitochondria after fertilization could potentially initiate an apoptotic cascade, and specifically, this cascade could be used to target spermatozoa structures through apoptotic pathways. However, such activation of apoptosis could spell disaster for the zygote and instead FUNDC2 delivered by spermatozoon could in fact play an undescribed role in mitophagy and perhaps initiate some step of sperm mitophagy after fertilization. Finally, FUNDC2 may just be a carryover protein from spermatogenesis which is detected by ooplasmic mitophagy machinery on the outer mitochondrial membrane and undergoes degradation along with the other proteins present there during post-fertilization sperm mitophagy.

SAMM50 is imbedded in the outer mitochondrial membrane and is a subunit of the sorting and assembly machinery (SAM) of the outer mitochondrial membrane. The SAM complex is responsible for the integration of proteins into the mitochondrial outer membrane (*Ott et al., 2012*). SAMM50 is also crucial for cristae structure, mitochondrial morphology, respiratory complex biogenesis, and mtDNA maintenance (*Ott et al., 2012*; *Xie et al., 2007*; *Liu et al., 2016*). SAMM50 was further investigated because of its role as a regulator of the PINK1-Parkin mitophagic pathway. PINK1-Parkin-mediated mitophagy is a well-characterized pathway in somatic cells where it regulates ubiquitin-dependent mitophagy and degrades defective mitochondria after mitochondrial damage takes place (*Palikaras et al., 2018*). A depletion of SAMM50 has been shown to result in PINK1 buildup, Parkin recruitment, and ultimately an increase in mitophagy via SQSTM1 (*Jian et al., 2018*). Such observations in somatic cells align with our studies, where the observed depletion of SAMM50 from extract-exposed spermatozoa may allow PINK1-Parkin-mediated mitophagy to degrade the sperm mitochondria upon entry into the oocyte cytoplasm. SAMM50 removal may be related to a role in prepping the mitochondrial sheath for degradation through the PINK1-Parkin mitophagy axis and may also involve proteasomal degradation. PINK1-Parkin mitophagy is a well-known somatic cell mitophagy pathway; however, it is yet to be documented to play a role in the zygotic post-fertilization sperm mitophagy. SAMM50 sperm dynamics may be a link between PINK1-Parkin mitophagy and post-fertilization mitophagy. SAMM50 has been shown to also interact directly with SQSTM1 and acts as a receptor for basal mitophagy of the MICOS complexes and the other components of the SAM complex (*Abudu et al., 2021*). Basal mitophagy is a housekeeping process in somatic cells rather than a stress response like the PINK1-Parkin mitophagic axis and can result in mitochondrial protein turnover rather than the degradation of entire mitochondria. During basal mitophagy, SAMM50 interacts with SQSTM1 to remove specific MICOS and SAM components and then actively replaces them. However, during post-fertilization mitophagy perhaps the SAMM50 which can still be observed on the fertilizing spermatozoa in zygotes (*Figure 7F and G*) is working in concert with SQSTM1 to degrade these key mitochondrial complexes but is unable to replace them as programmed post-fertilization sperm mitophagy continues. The reduction in SAMM50 from ooplasm-exposed sperm mitochondria may also be related to a role in prepping the mitochondrial sheath and destabilizing mitochondrial membrane for degradation through the PINK1-Parkin mitophagy axis and/or VCP-proteasome based proteolysis. When observing SAMM50 using immunocytochemistry, it was detected in the acrosome and mitochondrial sheath of ejaculated spermatozoa (*Figure 7B*). After priming, SAMM50 was still detected in the remnants of the acrosome and the mitochondrial sheath (*Figure 7C*), with the strongest signal present in the acrosome. After 4 hr of co-incubation within the cell-free system, the acrosomal localization of SAMM50

faded and the mitochondrial sheath localization remained (*Figure 7D*). A similar labeling pattern was again observed after 24 hr of cell-free exposure (*Figure 7E*). Our ICC results show that after cell-free treatment, all the remaining SAMM50 appears to be present in the mitochondrial sheath. However, the reduction of acrosomal SAMM50 content likely accounts for the decrease observed during our mass spectrometry trial. This acrosomal content is surprising considering SAMM50's defined role as a mitochondrial protein; however, SAMM50 may localize to membranes other than the mitochondrial outer membrane, including the acrosomal membranes. In zygotes 15 hr after insemination, SAMM50 appeared to cluster near the mitochondrial sheath of the fertilizing spermatozoa as well as on the mitochondrial sheath (*Figure 7F*); this clustering may be a combination of both oocyte-derived and sperm-derived SAMM50 being present near the mitochondrial sheath. Alternatively, this clustering may be SAMM50 molecules breaking off the mitochondrial sheath as the sheath begins to undergo degradation. In zygotes at 25 hr after insemination, SAMM50 is detected on the posterior portion of the mitochondrial sheath and the principal piece of the sperm tail (*Figure 7G*). SAMM50 may be acting as a pre-mitophagic regulator of the PINK1-Parkin mitophagic access here or perhaps it is helping to serve up mitochondria to a different mitophagic pathway (e.g. VCP-dependent proteasomal degradation). SAMM50 remains detectable on the mitochondrial sheath late into the one-cell stage. This SAMM50 may be serving up other mitochondrial proteins to SQSTM1, perhaps suggesting some type of dual mitophagic pathway role for SAMM50 during post-fertilization sperm mitophagy. SAMM50 is an integral mitochondrial component and may be more resistant to ooplasmic degradation factors than we would expect based on its known function as a mitophagy inhibitor.

BAG5 is another protein which is known to regulate the PINK1-Parkin mitophagic axis. BAG5 has been shown to interact with both Parkin and PINK1 (*Kalia et al., 2004*; *Wang et al., 2014*). BAG5 has been shown to inhibit Parkin activity (*Kalia et al., 2004*), by suppressing Parkin recruitment to damaged mitochondria and reducing the docking of damaged, autophagophore-enclosed mitochondria to lysosomes (*De Snoo et al., 2019*). This suppression causes an increase in the parkin-mediated degradation of the apoptosis-regulating protein, myeloid cell leukemia 1 (MCL1), which leads to cell death (*De Snoo et al., 2019*). BAG5 was also shown to inhibit PINK1 degradation (*Wang et al., 2014*). BAG5 appears to act as a sensor of cellular stress and regulating switch of PINK1-Parkin mediated mitophagy. In response to cellular stress, BAG5 can either be removed to allow selective mitophagy to take place, or BAG5 can remain, prevent Parkin recruitment, and instead cause apoptosis. Thus, BAG5's response appears to be dependent on the type and level of cellular stress experienced (*De Snoo et al., 2019*). The factors which mediate the effects of BAG5 on Parkin and PINK1 remain unknown. In the context of post-fertilization sperm mitophagy, the pro-selective mitophagy BAG5 regulated pathway would clearly be favored. The early removal of BAG5 from the fertilizing spermatozoa is likely promoting post-fertilization sperm mitophagy.

By ICC, we observed that BAG5 was exclusively located in the acrosome of freshly ejaculated spermatozoa (*Figure 8B*). After priming in the cell-free system, BAG5 continued to be localized in the acrosome with a faint signature appearing in the mitochondrial sheath (*Figure 8C*). After 4 and 24 hr of oocyte extract exposure, the BAG5 labeling was observed to be congregated on the mitochondrial sheath (*Figure 8D and E*). The majority of the BAG5 signal was in the acrosome of the spermatozoa and when the remains of the acrosome were removed by ooplasmic factors during cell-free system co-incubation, the BAG5 signal and content diminished. Following this same pattern, there was no clear localization pattern of BAG5 observed on the fertilizing spermatozoa in the IVF zygotes after either 15 or 25 hr (*Figure 8F and G*), the majority of BAG5 appears to have been removed from the fertilizing spermatozoa. This reduction of BAG5 content, which was observed in both the cell-free system and via IVF, may be promoting post-fertilization sperm mitophagy, in a fashion similar to somatic cell BAG5-regulated mitophagy.

In summary, our study harnessed comparative proteomic analysis in conjunction with our novel porcine cell-free system to capture proteomic alterations to spermatozoa which take place during oocyte cytoplasm exposure, a system which has been shown to faithfully mimic some early fertilization events. In total, 185 proteins were identified to undergo significant changes in abundance. Six of these proteins were further investigated and their localization patterns (*Figure 9*) were characterized in the porcine cell-free system and porcine zygotes Figure. These six proteins warrant further exploration but were able to showcase that our high-resolution mass spectrometry data can be replicated and further understood using immunocytochemistry. More work must be done to further understand these

six proteins and more candidates from the 185 proteins inventory should be investigated. However, this mass spectrometry study is the first of its kind to be conducted, and along with the further exploration of six candidates, it highlights the usefulness of our porcine cell-free system for the exploration of early fertilization events, such as post-fertilization sperm mitophagy. This novel system will remain a useful tool to explore early fertilization events at the molecular level.

## Materials and methods

### Antibodies and probes

Rabbit polyclonal anti-FUNDC2 (PA570823), MitoTracker Red CMXRos, and 4′,6-diamidino-2-phenylindole (DAPI) were purchased from Invitrogen, Waltham, MA. Mouse monoclonal anti-PSMA3 (BML-PW8115) was purchased from Enzo Life Sciences, Farmingdale, NY. Rabbit polyclonal anti-SAMM50 (20824-I-AP) was purchased from ProteinTech Group, Rosemount, IL. Mouse monoclonal anti-BAG5 (CF810618) was purchased from OriGene Technologies, Rockville, MD. Rabbit polyclonal anti-PACRG (ab4090), rabbit polyclonal anti-SPATA18 (180154), mouse monoclonal anti-MVP (ab14562), and rabbit polyclonal anti-PSMG2 (ab172909) were purchased from Abcam, Cambridge, United Kingdom. HRP-conjugated goat anti-mouse IgG (31430), goat anti-rabbit IgG (31460), goat anti-mouse IgG TRITC (T2762), goat anti-rabbit IgG FITC (65–6111), goat anti-rabbit IgG TRITC (T2769), and goat anti-mouse FITC (62–6511) were purchased from ThermoFischer Scientific, Waltham, MA. Beltsville thawing solution (BTS) boar semen extender supplied with gentamicin was purchased from IMV Technologies, L'Aigle, France. Unless otherwise noted, all chemicals used in this study were purchased from Sigma-Aldrich, St. Lious, MO.

### Boar semen collection and processing

Boars were housed at the University of Missouri Animal Science Research Center. Fresh boar semen was collected in one regular collection per week, transferred into 15 mL centrifuge tubes, and centrifuged at $800 \times g$ for 10 min to separate spermatozoa from seminal plasma. Sperm concentration was assessed by using a light microscope and a hemocytometer (Thermo Fischer Scientific, Waltham, MA). Only semen collections with >80% motile spermatozoa and <20% morphological abnormalities were used. Spermatozoa were diluted with BTS to a final concentration of $1 \times 10^8$ spermatozoa/mL and stored in a sperm incubator at 17 °C for up to 5 days.

### Collection and in vitro maturation (IVM) of pig oocytes

Porcine ovaries were obtained from a local slaughterhouse. Cumulus-oocyte complexes (COCs) were aspirated from antral follicles of 2–6 mm size and washed three times with HEPES buffered Tyrode Lactate medium containing 0.01% (w/v) polyvinyl alcohol (TL-HEPES-PVA). COCs were transferred into 500 µL wells of oocyte maturation medium (TCM 199, Mediatech, Inc, Manassas, VA) supplemented with 0.1% PVA, 3.05 mM D-glucose, 0.91 mM sodium pyruvate, 20 µg/mL of gentamicin, 0.57 mM cysteine, 0.5 µg/mL LH (L5269), 0.5 µg/mL FSH (F2293), 10 ng/mL epidermal growth factor (E4127), 10% (v/v) porcine follicular fluid. The media was overlaid with mineral oil in four-well dishes (ThermoFischer) and the COCs were incubated at 38.5 °C, with 5% $CO_2$ in the air, for 40–44 hours.

### In vitro fertilization (IVF) and in vitro culture (IVC) of pig oocytes/ zygotes

Cumulus cells of matured COCs were removed with 0.1% (w/v) hyaluronidase in TL-HEPES-PVA medium. MII oocytes as identified by the presence of a polar body. Mature oocytes were then washed three times with TL-HEPES-PVA medium and once with Tris-buffered medium (mTBM) containing 0.3% (w/v) BSA (A7888). Between 30–40 oocytes were placed into 100 µL drops of the mTBM covered with mineral oil in a 35 mm polystyrene culture dish, then incubated until spermatozoa were prepared for fertilization. Liquid semen preserved in BTS extender solution was washed with PBS containing 0.1% (w/v) PVA (PBS-PVA) two times by centrifugation at $800 \times g$ for 5 min. To stain mitochondria in the sperm tail, the boar spermatozoa were incubated with vital, fixable, mitochondrion-specific probe MitoTracker Red CMXRos for 10 min at 38.5 °C. The spermatozoa pre-labeled with MitoTracker were resuspended in mTBM medium. The sperm suspension in mTBM medium was then added to the 100 µL drops of mTBM medium for a final concentration of $2.5–5 \times 10^5$ spermatozoa/mL. Matured

oocytes were incubated with spermatozoa for 5 hours at 38.5 °C, with 5% $CO_2$ in the air, then transferred to 500 μL drops of MU3 medium [58] containing 0.4% (w/v) BSA (A6003) for additional culture.

## Sperm priming for cell-free system

Boar spermatozoa were washed with phosphate-buffered saline (PBS, 137 mM NaCl, 2.7 mM KCl, 10 mM $Na_2HPO_4$, 1.8 mM $KH_2PO_4$, pH = 7.2) containing 0.1% (w/v) PVA (PBS-PVA) two times by centrifugation at 800×*g* for 5 min. The sperm mitochondria were labeled with MitoTracker Red CMXRos for 10 min at 37 °C. At the previously tested concentration of 400 nM, the probe specifically stains boar sperm mitochondria but is also taken up by the sperm head structures (*Song et al., 2016*).

To prime sperm mitochondrial sheaths for cell-free studies, spermatozoa pre-labeled with MitoTracker were demembranated/permeabilized with 0.05% (w/v) L-α-lysophosphatidylcholine in KMT (20 mM KCl, 5 mM $MgCl_2$, 50 mM TRIS·HCl, pH = 7.0) for 10 min at 37 °C and washed twice with the KMT for 5 min by centrifugation, to terminate the reaction. The spermatozoa were subsequently incubated with 1.0 mM dithiothreitol (DTT) diluted in KMT, pH = 8.2 for 20 min at 37 °C and washed twice with KMT for 5 min by centrifugation, to terminate the reaction.

## Preparation of porcine oocyte extracts

Cumulus cells of matured COCs were removed with 0.1% (w/v) hyaluronidase in TL-HEPES-PVA medium. The oocytes were then searched for mature MII oocytes as designated by the presence of a polar body. Mature oocytes were then washed three times with TL-HEPES-PVA medium. Zonae pellucidae (ZP) were removed by 0.1% (w/v) pronase (Sigma) in TL-HEPES-PVA. The ZP-free, mature MII oocytes were transferred into an extraction buffer (50 mM KCl, 5 mM $MgCl_2$, 5 mM ethylene glycol-bis[β-aminoethyl ether]-N,N,N′,N′-tetraacetic acid [EGTA], 2 mM β-mercaptoethanol, 0.1 mM PMSF, protease inhibitor cocktail [78410, ThermoFischer Scientific], 50 mM HEPES, pH = 7.6) containing an energy-regenerating system (2 mM ATP, 20 mM phosphocreatine, 20 U/mL creatine kinase, and 2 mM GTP), and submerged three times into liquid nitrogen for 5 min each. Next, the frozen-thawed oocytes were crushed by high-speed centrifugation at 16,650×*g* for 20 min at 4 °C in a Sorvall Biofuge Fresco (Kendro Laboratory Products). Batches of oocyte extract were made from 1000 oocytes in 100 μL of extract. The supernatants were harvested, transferred into a 1.5 mL tube, and stored in a deep freezer (–80 °C).

## Co-Incubation of ermeabilized mammalian spermatozoa with porcine oocyte extracts

The permeabilized boar spermatozoa were added to porcine oocyte extracts at a concentration of 1x$10^4$ spermatozoa/10 μL of an extract and co-incubated for 4–24 hr in an incubator at 38.5 °C, with 5% $CO_2$ in the air. After co-incubation, spermatozoa were washed 3 x with KMT. At which point the spermatozoa were either processed for immunocytochemistry (as described below), or prepared for mass spectrometry analysis (as described below).

## Immunocytochemistry

The ICC protocol was performed as described previously (*Sutovsky et al., 2004*). Briefly, to fix oocytes or embryos, cells were fixed with 2% (v/v) formaldehyde in PBS for 40 min at room temperature, washed and processed, or stored in PBS at 4 °C until used for immunocytochemistry. In some cases, for oocytes and embryos, zona pellucida was removed using 0.1%, (w/v) pronase in TL-HEPES-PVA and the cells were then fixed with 2% (v/v) formaldehyde in PBS for 40 min at room temp. To fix ejaculated, primed, or cell-free-treated boar spermatozoa; microscopy coverslips were overlaid with 300 μL of 1% (w/v) aqueous solution of poly-L-lysine, incubated for 5 min, then shaken off, and allowed to dry. The poly-L-lysin coated coverslips were overlaid with 400 μL of warm KMT medium (37 °C; pH = 7.3) and 2 μL of sperm suspension (1×$10^8$ spermatozoa/mL) were added onto coverslips and allowed to settle on the lysine-coated surface for 10 min on a 38.5 °C plate. KMT was shaken off from the coverslips and coverslips were overlaid with 2% (v/v) formaldehyde in PBS for 40 min fixation at room temperature, then used for immunocytochemistry immediately or stored at 4 °C. Both spermatozoa and oocytes were permeabilized in PBS with 0.1% (v/v) Triton X-100 (PBST) at room temperature for 40 min, then blocked with 5% (v/v) normal goat serum (NGS) in PBST for 25 min. Spermatozoa/oocytes were incubated with the appropriate primary antibodies diluted in PBST containing 1% (v/v)

NGS overnight at 4 °C. The primary antibodies used throughout the trial and their dilution ratios are as follows: anti-FUNDC2 (1:50), anti-PACRG (1:50), anti-SPATA18 (1:50), anti-MVP (1:50), anti-SAMM50 (1:50), anti-BAG5 (1:50), anti-PSMG2 (1:25), and anti-PSMA3 (1:25). The samples were incubated with appropriate species-specific secondary antibodies such as goat-anti-rabbit (GAR)-IgG-FITC (1:100 dilution), GAR-IgG-TRITC (1:100), goat-anti-mouse (GAM)-IgG-FITC (1:100), or GAM-IgG-TRITC, all diluted 1:100 in PBST with 1% (v/v) NGS for 40 min at room temperature; 2.5 µg/mL DNA stain DAPI was included as well. The samples were mounted on microscopy slides in VectaShield mounting medium (Vector Laboratories, Burlingame, CA), and imaged using a Nikon Eclipse 800 microscope (Nikon Instruments Inc, Melville, NY) with a Retiga QI-R6 camera (Teledyne QImaging, Surrey, BC, Canada) operated by MetaMorph 7.10.2.240. software (Molecular Devices, San Jose, CA). For each immunocytochemistry replicate, either 200 spermatozoa or 20 zygotes were counted. For all images presented, a picture capturing the typical localization pattern is displayed, representative of three replicates of immunolabelling per image. The displayed patterns were observed between 65 to 88% of observed and counted spermatozoa/zygotes. Images were adjusted for contrast and brightness in Adobe Photoshop 2020 (Adobe Systems, Mountain View, CA), to match the fluorescence intensities viewed through the microscope eyepieces.

## SDS-PAGE and western blotting

The oocyte extract proteins were prepared for WB by mixing the oocyte extract with 4×LDS loading buffer and ultrapure water to obtain 1×LDS (106 mM Tris·HCl, 141 mM Tris base, 2% (w/v) LDS, 10% (w/v) Glycerol, 0.75% (w/v) Coomassie Blue G250, 0.025% (w/v) Phenol Red, pH = 8.5), supplemented with 2.5% (v/v) β-mercaptoethanol and incubated at 70 °C for 10 min prior to use. Spermatozoa were suspended in 1×LDS loading buffer supplemented with 2.5% β-mercaptoethanol. Spermatozoa were incubated at room temperature in 1×LDS loading buffer supplemented with protease inhibitor cocktail for 1 hr on a rocking platform and spun. Total protein equivalent of 10–20 million spermatozoa, and 100 MII oocytes were loaded per lane, respectively, on a NuPAGE 4–12% Bis-Tris gel (Invitrogen). Electrophoresis was carried out in the Bis-Tris system using MOPS-SDS running buffer (50 mM MOPS, 50 mM Tris base, 0.1% [w/v] SDS, 1 mM EDTA, pH = 7.7) with the cathode buffer supplemented with 5 mM sodium bisulfite. The molecular masses of separated proteins were estimated using Novex Sharp Pre-stained Protein Standard (Invitrogen, LC5800) run in parallel. PAGE was carried out for 5 min at 90 Volts to let the samples delve into the gel and then for another 60–70 min at 200 Volts. The power was limited to 20 Watts. After PAGE, proteins were electro-transferred to polyvinylidene fluoride (PVDF) membranes (MilliporeSigma, Burlington, MA) using Owl wet transfer system (ThermoFischer Scientific) at 65 Volts for 90 min for immunodetection, using Bis-Tris-Bicine transfer buffer (25 mM Bis-Tris base, 25 mM Bicine, 1 mM EDTA, pH = 7.2) supplemented with 10% (v/v) methanol per membrane, and 2.5 mM sodium bisulfite. The membranes with the transferred proteins were blocked with 10% (w/v) non-fat milk in TBS with 0.05% (v/v) Tween 20 (TBST, Sigma) for 40 min. The membranes were then incubated with the appropriate primary antibodies diluted in 5% non-fat milk in TBST overnight at 4 °C. The primary antibodies used throughout the trial and their dilution ratios are as follows: anti-FUNDC2 (1:1000), anti-PACRG (1:1000), anti-SPATA18 (1:1000), anti-MVP (1:1000), anti-SAMM50 (1:1000), anti-BAG5 (1:1000), anti-PSMG2 (1:1000) and anti-PSMA3 (1:1000). The membranes were subsequently incubated with appropriate species-specific secondary antibodies such as HRP-conjugated goat anti-mouse IgG (GAM-IgG-HRP), or goat anti-rabbit (GAR-IgG-HRP) for 40 min at room temperature. The membranes were reacted with chemiluminescent substrate (Millipore), detected using ChemiDoc Touch Imaging System (Bio-Rad, Hercules, CA, USA) to record the protein bands, and analyzed by Image Lab Software (ver. 5.2.1, Bio-Rad, Hercules, CA, USA). The membranes were stained with CBB R-250 after chemiluminescence detection for protein load control.

## Mass spectrometry sample preparation

Cell-free system exposed spermatozoa, spermatozoa controls, and oocyte extract underwent protein precipitation using a TCA protein precipitation protocol from Dr. Luis Sanchez. These samples were then resuspended in acetone and submitted to the University of Missouri Gehrke Proteomics Center for MALDI-TOF Mass Spectrometry analysis. At the Proteomics Center, these samples were washed with 80% cold acetone twice. Then 10 µl 6 M urea 2 M thiourea and 100 mM ammonium bicarbonate was added to the protein pellet. Solubilized protein was reduced by DTT and alkylated by

iodoacetamide. Then trypsin was added for digestion overnight. The digested peptides were C18 ZipTip desalted, lyophilized, and resuspended in 10 µL 5/0.1% acetonitrile/formic acid.

A volume of 1 µL of suspended peptides was loaded onto a C18 column with a step gradient of acetonitrile at 300 nL/min. A Bruker nanoElute system was connected to a timsTOF pro mass spectrometer. The loaded peptide was eluted at a flow rate of 300 nL/min with the initial gradient of 3% B (A: 0.1% formic acid in water, B: 99.9% acetonitrile, 0.1% formic acid), followed by 11 min ramp to 17%B, 17–25% B over 21 min, 25–37% B over 10 min, 37–80% B over 4 min, holding at 80% B for 9 min, 80–3% B in 1 min, and holding at 3% B for 3 min. The total running time was 60 min.

Raw data was searched using PEAKs (version X+) with UniProt *Sus scrofa* protein database downloaded on March 01, 2019, with 88,374 entries. Samples were adjusted for trypsin digestion, 4 missed cleavages allowed; carbamidomethyl cysteine as a fixed modification; oxidized methionine and acetylation on protein N terminus as variable modification. 50 ppm mass tolerance on precursor ions, 0.1 Da on fragment ions. For protein identification, the following criteria were used: peptide FDR and protein FDR <1%, and ≥ 4 spectra per protein in each sample. Samples were submitted in triplicate for both the 4- and 24 hr cell-free system trials.

## Mass spectrometry data statistical analysis

Prior to statistical analysis, the primed and cell-free treated sperm samples were normalized based on the content of outer dense fiber proteins (ODF) 1, 2, and 3. To further reduce batch variance, the protein spectrum counts were also subject to normalization by means. After these normalization steps, the primed and cell-free extract-treated sperm samples were statistically compared using a paired T-test. This T-test was comparing the relative normalized protein abundance between our primed control and cell-free treated samples. $p < 0.1$ was considered statistical significance for Class 2 and Class 3 proteins. For Class 1 proteins $p < 0.2$ was considered statistically significant. The statistical parameters were loosened slightly for Class 1 proteins because proteins in this classification were only found to be present after cell-free system exposure. Whether these proteins were determined significant via statistical analysis or not, we assume that the observed distribution patter changes warrant addition to our candidate list even if they don not fall under a more typical $p < 0.1$ statistical threshold.

## Protein Classification

Both the 4 hr and 24 hr protein inventories were divided into three different classes. Class 1 proteins were detected only in the oocyte extract (not in the vehicle control or primed control spermatozoa) and found on the spermatozoa only after extract co-incubation. These proteins are interpreted as ooplasmic mitophagy receptors/determinants and nuclear/centrosomal remodeling factors ($p < 0.2$). Class 2 proteins were detected in the primed spermatozoa but increased in the spermatozoa exposed to cell-free system co-incubation ($p < 0.1$). Class 3 proteins were present in both the gametes or only the spermatozoa, but are decreased in the spermatozoa after co-incubation, interpreted as sperm-borne mitophagy determinants and/or sperm-borne proteolytic substrates of the oocyte autophagic system ($p < 0.1$).

## Functional analysis

Following the statistical analysis and protein classification, the functions of all proteins $p < 0.1$ ($< 0.2$ for Class 1), were searched using a PubMed literature search and UniProt Knowledgebase search. Known functions can be found in *Supplementary files 2 and 3*. Proteins were then categorized based on known functions and known roles within pathways, in gametes and/or somatic cells. Protein categorization results can be found in the pie chart images of *Figure 2*.

## Acknowledgements

We are truly thankful for the support received from the staff of the National Swine Research and Resource Center, the University of Missouri, funded by National Institutes of Health (NIH) grant U42 OD011140, as well as Professor Randall Prather and his associates for their kind support, including but not limited to gilt ovary and boar semen collections. We would also like to thank the University of Missouri Gehrke Proteomics Center, and its staff who conducted the mass spectrometry data

collection and statistical analysis. Study was funded by USDA-NIFA grant number 2020-67015-31017 and seed funding from the College of Agriculture, Food and Natural Resources, University of Missouri.

## Additional information

### Funding

| Funder | Grant reference number | Author |
|---|---|---|
| National Institute of Food and Agriculture | 2020-67015-31017 | Peter Sutovsky |

The funders had no role in study design, data collection and interpretation, or the decision to submit the work for publication.

### Author contributions
Dalen Zuidema, Conceptualization, Data curation, Formal analysis, Validation, Investigation, Visualization, Methodology, Writing – original draft, Writing – review and editing; Alexis Jones, Formal analysis, Investigation, Visualization, Writing – original draft; Won-Hee Song, Data curation, Formal analysis, Validation, Visualization, Methodology, Writing – original draft; Michal Zigo, Data curation, Formal analysis, Validation, Investigation, Visualization, Methodology, Writing – original draft; Peter Sutovsky, Conceptualization, Resources, Data curation, Formal analysis, Supervision, Funding acquisition, Investigation, Methodology, Writing – original draft, Project administration, Writing – review and editing

### Author ORCIDs
Dalen Zuidema http://orcid.org/0000-0002-5363-0407
Michal Zigo http://orcid.org/0000-0002-2964-5940
Peter Sutovsky http://orcid.org/0000-0002-9231-2823

Reviewer #1 (Public Review): https://doi.org/10.7554/eLife.85596.3.sa1
Reviewer #2 (Public Review): https://doi.org/10.7554/eLife.85596.3.sa2
Reviewer #3 (Public Review): https://doi.org/10.7554/eLife.85596.3.sa3
Author Response https://doi.org/10.7554/eLife.85596.3.sa4

## Additional files

### Supplementary files
• Supplementary file 1. Raw data captured from mass spectrometry, referenced against the *Sus scrofa* UniProt Knowledge base.

• Supplementary file 2. Proteomic identification of mitophagy and sperm remodeling cofactors in the porcine cell-free system after 4 hours of co-incubation.

• Supplementary file 3. Proteomic identification of mitophagy and sperm remodeling cofactors in porcine cell-free system after 24 hours of co-incubation.

• MDAR checklist

### Data availability
Porcine cell-free system mass spectrometry compiled data sets are available on Dryad (https://doi.org/10.5061/dryad.t4b8gtj5v).

The following dataset was generated:

| Author(s) | Year | Dataset title | Dataset URL | Database and Identifier |
|---|---|---|---|---|
| Zuidema D, Sutovsky P | 2023 | Porcine cell-free system mass spectrometry compiled data sets | https://doi.org/10.5061/dryad.t4b8gtj5v | Dryad Digital Repository, 10.5061/dryad.t4b8gtj5v |

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
