## [Editor Report · eLife assessment]

This **important** work reports the identification of a list of proteins that may participate in the clearance of paternal mitochondria during fertilization, which is known as essential for normal fertilization and embryonic and fetal development. The main method used is state-of-the-art and the supporting data are **solid**. This work will be of interest to developmental and reproductive biologists working on fertilization.

---

## [Referee Report · Reviewer #1 (Public Review)]

In this manuscript, the authors used an unbiased method to identify proteins from porcine oocyte extracts associated with permeabilised boar spermatozoa in vitro. The identification of the proteins is done by mass spectrometry. A previous publication of this lab validated the cell-free extract purification methods as recapitulating early events after sperm entry in the oocyte. This novel method with mammalian gametes has the advantage that it can be done with many spermatozoa at the time and allows the identification of proteins associated with many permeabilised boar spermatozoa at the time. This allowed the authors to establish a list of proteins either enriched or depleted after incubation with the oocytes extract or even only associated with spermatozoa after incubation for 4h or 24h. The total number of proteins identified in their test is around 200 and with very few present in the sample only when spermatozoa were incubated with the extracts.

The list of proteins identified using this approach and these criteria provide a list of proteins likely associated with spermatozoa remnants after their entry and either removed or recruited for the transformation of spermatozoa-derived structures. Using WB and histochemistry labelling of spermatozoa and early embryos using specific antibodies the authors confirmed the association/dissociation of 6 proteins suspected to be involved in autophagy.

While this unique approach provides a list of potential proteins involved in sperm mitochondria clearance it is (only) a starting point for many future studies and does not provide the demonstration that any of these proteins has indeed a role in the processes leading to sperm mitochondria clearance. The protein identified may also be involved in other processes going on in the oocyte at this time of early development.

---

## [Referee Report · Reviewer #2 (Public Review)]

Mitochondria are essential cellular organelles that generate ATPs as the energy source for maintaining regular cellular functions. However, the degradation of sperm-borne mitochondria after fertilization is a conserved event known as mitophagy to ensure the exclusively maternal inheritance of the mitochondrial DNA genome. Defects on post-fertilization sperm mitophagy will lead to fatal consequences in patients. Therefore, understanding the cellular and molecular regulation of the post-fertilization sperm mitophagy process is critically important. In this study, Zuidema et. al applied mass spectrometry in conjunction with a porcine cell-free system to identify potential autophagic cofactors involved in post-fertilization sperm mitophagy. They identified a list of 185 proteins that might be candidates for mitophagy determinants (or their co-factors). Despite the fact that 6 (out of 185) proteins were further studied, based on their known functions, using a porcine cell-free system in conjunction with immunocytochemistry and Western blotting, to characterize the localization and modification changes these proteins, no further functional validation experiments were performed. Nevertheless, the data presented in the current study is of great interest and could be important for future studies in this field.

---

## [Referee Report · Reviewer #3 (Public Review)]

In this manuscript, a cytosolic extract of porcine oocytes is prepared. To this end, the authors have aspirated follicles from ovaries obtained by first maturing oocytes to meiose 2 metaphase stage (one polar body) from the slaughterhouse. Cumulus cells (hyaluronidase treatment) and the zona pellucida (pronase treatment) were removed and the resulting naked mature oocytes (1000 per portion) were extracted in a buffer containing divalent cation chelator, beta-mercaptoethanol, protease inhibitors, and a creatine kinase phosphocreatine cocktail for energy regeneration which was subsequently triple frozen/thawed in liquid nitrogen and crushed by 16 kG centrifugation. The supernatant (1.5 mL) was harvested and 10 microliters of it were used for interaction with 10,000 permeabilized boar sperm per 10 microliter extract (which thus represents the cytosol fraction of 6.67 oocytes).

The sperm were in this assay treated with DTT and lysoPC to prime the sperm's mitochondrial sheath.

After incubation and washing these preps were used for Western blot for Fluorescence microscopy and for proteomic identification of proteins. I am very positive about the porcine cell-free assay approach and the results presented here.

---

## [Author Response]

The following is the authors’ response to the original reviews.

This important work reports the identification of a list of proteins that may participate in the clearance of paternal mitochondria during fertilization, which is known as essential for normal fertilization and embryonic and fetal development. While the main method used is state of the art and the supporting data are solid, the vigor of the biochemical assays and function validation is inadequate. This work will be of interest to developmental and reproductive biologists working on fertilization. Key revisions (for the authors) include (1) Use a mitochondria-enriched fraction instead of whole sperm for the assays, and add more control samples to monitor what got lost during sperm and oocyte treatments before the coincubation step. (2) Functional validation of the key proteins identified.

We thank Editors of eLife, as well as Special Issue Guest-Editors and Reviewers for a favorable assessment and helpful recommendations for key revisions. Provisional revisions included in our revised article are detailed below. We agree with Editors’ comment about the use of mitochondrion enriched fractions and additional functional validation of key proteins. In fact, we are developing experimental protocols for oocyte extract coincubation with isolated sperm heads and tails, and eventually with purified mitochondrial sheaths, to separate the ooplasmic sperm nucleus remodeling factors from the mitophagic ones. Such experiments, as well as functional validations using porcine zygotes are contingent upon anticipated post-pandemic rebound in the availability of porcine oocytes, obtained from ovaries harvested on slaughterhouse floors, requiring currently unavailable workforce which has hampered our access to this necessary resource.

**Reviewer #1 (Peer Review):**
Could the authors make clear how much the presented pictures reflect the described localisation? There is no information on the number of spermatozoa and embryos observed nor the fraction of these embryos showing the presented pattern of localisation. This must be included.

Two hundred spermatozoa were counted per replicate of the cell-free system co-incubation and 20 zygotes per replicate, with 3 replicates of immunolabelling for each phase/picture which were examined to establish the typical localization patterns that were observed. The displayed patterns were observed in 65 to 88% of examined spermatozoa/zygotes; varying dependent on protein, replicate, and phase of immunolabelling. In all cases, the signal displayed is the typical pattern that was displayed in most cells. This information has been added to the Materials and Methods section for clarification.

It is not clear if the authors also examined the localization of other proteins and obtained a different pattern than anticipated from the proteomic approach or if they only tested these 6 proteins and got a 100% of correlation.

These are the 6 proteins which were selected based on extensive literature review into known functions of all identified proteins, as well as extensive research into available and reliable antibodies to detect such proteins within our porcine systems. Even so, no particular localization patterns were anticipated; instead, we presented the patterns actually observed and even some patterns which defied our expectations (i.e., the localization of BAG5 in the sperm acrosome).

The authors use "MS" in the text to indicate "mitochondrial Sheath" and "Mass spectrometry". this is confusing.

The authors agree and the usage of MS as an acronym for either has been removed entirely to avoid confusion.

In the introduction the author refers to Ankel-Simons and Cummins, 1996 as a reference for the number of sperm mitochondria in mammalian species, this is incorrect since the quoted paper is about the number of mtDNA molecules and mentioned an earlier publication.

This has been revised and the appropriate citation has been used.

**Reviewer #2 (Peer Review):**
Major:1. It has been proved from the earlier studies from this group that the porcine cell-free system is useful to observe spermatozoa interacting with ooplasmic proteins in a single trial and could recapitulate fertilization sperm mitophagy events that take place in a zygote without affecting later cell-division process. However, the post-fertilization sperm mitophagy process is a complex time-associated event that many processes that occur sequentially and interactively, which means ooplasmic proteins might be involved in this process but may not directly interact with sperm or may associate with sperm-ooplasmic protein complex at different time points. It is certainly a great advance already in knowledge to identify "the candidate players" from the list of 185 proteins; however, with the time-resolution (4 and 24hr) in the current study and without functional validation experiments at this stage, it is still difficult to postulate the importance of these identified proteins. The functional validation experimental designs, in my opinion, is critically important for better interpretation of the data.

The authors agree with this reviewer’s sentiments and do plan to conduct further functional analysis. This project was able to generate a list of candidate, sperm-mitophagy promoting proteins and we were further able to show that many of these proteins were detectable both via mass spectrometry and via immunocytochemistry in spermatozoa exposed to our cell-free system. Furthermore, similar localization patterns were found in spermatozoa that were detected within newly fertilized zygotes. These results boost our confidence in our cell-free system and show that our list of candidate proteins is truly a useful list for future localization and functional analyses. We are certainly aware that we have not captured every protein that may play a role in post-fertilization sperm mitophagy and that the proteins captured are just candidates until proven otherwise. Likewise, we have almost certainly captured multiple proteins that are currently candidates that will likely not be shown to play a role in postfertilization sperm mitophagy, while it is plausible that at least some of these candidate proteins do play a role in mitophagy and some of them likely participate (perhaps have yet to be described roles) in other fertilization events, in which we would be extremely interested in as well.

1. As shown in Figure 1, whole sperm was used in the co-incubation and the later MS analysis; thus, proteins identified in the current study might be relevant in fertilization processes other than postfertilization sperm mitophagy, as proteins identified in the current study may be associated with other parts of the sperm (e.g. sticky sperm head, e.g. PSMG2 associated with sperm midpieces, tail at 4hr coincubation, but then only associate with sperm head at 24hr co-incubation) rather than sperm midpiece, despite the fact that authors applied immunohistochemistry to show the localization of this protein, but the evidence is indirect, so how authors functionally differentiate these 6 identified proteins from sperm mitophagy process with other processes and to confirm (or to associate) the relevance of these proteins with sperm mitophagy process?

The authors agree that the 6 proteins which were further studied by using immunocytochemistry may be playing roles in other processes such as pronuclear formation. We discussed some potential roles including and beyond post-fertilization mitophagy, in the Supplemental Discussion. After reviewer comments, we moved the Supplemental Discussion back in the main Discussion section. Thus, this section now considers additional putative pathways in which the said 6 proteins cold participate, though we concede that thorough functional studies must still be performed.

1. Class 3 proteins were present in both the gametes or only the primed control spermatozoa, but are decreased in the spermatozoa after co-incubation, which authors interpreted as sperm-borne mitophagy determinants and/or sperm-borne proteolytic substrates of the oocyte autophagic system, this data categorization may need to be revised as sperm-borne proteolytic substrates of the oocyte autophagic system only, not for sperm borne mitophagy determinants. The argument for this disagreement is due to the fact that if the protein is a sperm-borne mitophagy determinant, after coincubation, to execute the mitophagy process, this protein should still be associated with the sperm at least at the early stage (of 4hr) (constant under MS detection when comparing control with 4hr treated) rather than being released from the sperm. Or alternatively, they could result in class 3 proteins (but not all those 6 were in class 3). Nevertheless, if these proteins serve as substrates, they can be used (consumed) and show decreased under MS detection.

This argument for redefining the Class 3 proteins more accurately is understood and we agree. The definition is revised in the paper.

1. Of particular interest among the 6 proteins that were further investigated. Unlike other proteins, MVP was highly significant (p<0.001) after 4hr incubation, but the significance became less after 24hr (p=0.19). Interpretation of this dynamic change in the relevance of the mitophagy process would facilitate the readers to understand the relevance and the role of MVP.

The differences in significance are likely influenced by the abundance of MVP detectable by mass spectrometry. As the time of cell-free system incubation increases, the variability between replicates also seemed to increase, likely due to the sustained proteolytic activity taking place in our system. This work was based on three replicates of mass spectrometry for each time point; additional replicates likely would have reduced the p-value for the 24hr cell-free data set, for MVP and potentially other proteins also. At both time points, MVP was only detectable in spermatozoa after they had been exposed to the cell-free system treatment which is the criteria that truly interested us more than the actual differences in content between the timepoints and is why it was added to our list of candidate proteins.

1. In figure 3, the association of ooplasmic MVP to sperm midpiece is not convincing enough as sperm midpiece and tail often show some levels of non-specific signals under fluorescent microscopy. And the dynamic association of ooplasmic MVP to sperm midpiece in Fig. 3F-G is difficult to reach a conclusion solely based on data presented in the manuscript. Additional negative control of sperm MVP staining from the primed and treated sperm would be helpful. Additionally, a quantitative comparison (15 vs 25hr) of sperm-associated MVP signals from the fertilized embryo or a stack image from different angles would clarify the doubts raised here.

For all images and all replicates, serum controls were also generated. These controls were then viewed under fluorescent microscope, and light intensities and exposures thresholds for each fluorescent light channel were set based on the background intensity that came from these nonimmune serum-treated control samples. We set our light intensity/acquisition time below a threshold where the non-specific signal began to appear. All the presented patterns are based on setting this peak intensity threshold and as such the signal we see should be the true signal. Furthermore, 200 spermatozoa were counted per treatment per replicate of the cell-free system co-incubation and 20 zygotes per replicate, with 3 replicates of immunolabelling for each protein and data point, which was used to represent the typical localization patterns that were observed. The displayed patterns were observed between in 65- 88% of examined spermatozoa/zygotes. Invariably, the signal displayed in the manuscript is the typical pattern that was seen in a majority of cells. This information has now been added to the Materials & Methods section for clarification.

1. Same concerns for the other 5 proteins (PSMG2, PSMA3, FUNDC2, SAMM50, BAG5) as indicated above.

See response to Question 5.

1. The patterns of these 6 proteins under the immunofluorescent study are confusing as the pattern varies after co-incubation (treated), and mostly, the signal of these proteins observed from the fertilized embryos is not really associated with sperm midpieces. Therefore, the evidence of these proteins involving in post-fertilization sperm mitophagy is, at this moment, weak based on the data presented. But the relevance of these proteins in events post-fertilization or early embryo development is certainly (evidence did not strong enough to support "sperm mitophagy," in my opinion).

The authors agree that some of these proteins seem to be playing roles beyond postfertilization sperm mitophagy and that there is a need for true functional studies before the authors can state with certainty that these proteins play a role in any of the discussed fertilization events. We state this in the discussion: “Considering the dynamic proteomic remodeling of both the oocyte and spermatozoa which takes place during early fertilization, these 185 proteins which have been identified likely play roles in processes beyond sperm mitophagy.” It should be noted that the authors went into greater detail about potential alternative protein functions based on the present data and literature review in the Supplemental Discussion. Based on this comment and other reviewer comments we have now included the Supplemental Discussion as part of the main Discussion section, and this will hopefully help clarify some of the authors’ thoughts about the 6 candidate proteins which were further analyzed during this study.

Minor:1. To my understanding, statistical significance (relevance) is normally set at a p-value of either <0.1 or 0.05. The reason for loosening the p-value of 0.2 in the current study needs to be justified as this was not a common statistical criterium, and the interpretation of those candidates from this loosened criterium should also be careful.

The loosening of statistical relevance in this study to 0.2, only applied to our Class 1 proteins. This is because for a protein to fall into the Class 1 proteins it was a protein that was only present in samples after they were exposed to the cell-free system. In the case of these Class 1 proteins, this happened for all 3 replicates at each stated timepoint. We found this pattern of detection to be important whether the p-value fell under 0.1 or 0.2. As such, we loosened our statistical threshold for our Class 1 proteins. Any proteins added to our candidate list will be subject to further investigation before definitive conclusions can be drawn, and as such we think that capturing more proteins was more important for the goals of this study than limiting the number of proteins captured, especially for those Class 1 proteins. An explanation of this has been added to the Materials & Methods section Mass Spectrometry Data Statistical Analysis.

1. First cell cleavage of porcine embryo normally occurs within 48hr post-insemination or activation; therefore, the 4 and the 24hr time points used in the current study require justification included in the discussion or methods and material section.

First cleavage of porcine embryos normally occurs around 24 - 28 hours post-insemination. Thus, for both the cell-free system and the embryo studies we were capturing an advanced 1 cell stage zygote/zygote like system with our 24 hour and 25-hour time points.

1. In figure 2, colors used in different time points and in two different classes represent (sometimes) different protein categories, would be easier for the readers for quick comparisons if the same color could be used to represent the same protein category throughout the graph. (E.g, proteins for early zygote development are shown in red in "A", but blue in "B")

This has been corrected and the color scheme for Figure 2 has been revised for easier comparisons.

**Reviewer #3 (Peer Review):**
I am not used to seeing a supplementary discussion in a manuscript. I also believe it should be incorporated into normal discussion.

The Supplemental Discussion has been incorporated into the main Discussion now.

It would be very helpful to make an additional figure in which the proposed interactome of identified factors with the sperm mitochondria before and after incubation are drawn schematically and also which factors are not IDed in both cases (when comparing to somatic mito- or autophagy). This eases to get through the discussion and will beautifully summarize and illustrate the importance and progress that the authors have made with this assay.

We made a diagram that depicts the changes in protein localization patterns overtime within our cell-free system. This diagram has been added to the manuscript as Figure 9.

**Reviewer #1 (Public Review):**
In this manuscript, the authors used an unbiased method to identify proteins from porcine oocyte extracts associated with permeabilised boar spermatozoa in vitro. The identification of the proteins is done by mass spectrometry. A previous publication of this lab validated the cell-free extract purification methods as recapitulating early events after sperm entry in the oocyte. This novel method with mammalian gametes has the advantage that it can be done with many spermatozoa at the time and allows the identification of proteins associated with many permeabilised boar spermatozoa at the time. This allowed the authors to establish a list of proteins either enriched or depleted after incubation with the oocytes extract or even only associated with spermatozoa after incubation for 4h or 24h. The total number of proteins identified in their test is around 2 hundred and with very few present in the sample only when spermatozoa were incubated with the extracts. The list of proteins identified using this approach and these criteria provide a list of proteins likely associated with spermatozoa remnants after their entry and either removed or recruited for the transformation of spermatozoa-derived structures. Using WB and histochemistry labelling of spermatozoa and early embryos using specific antibodies the authors confirmed the association/dissociation of 6 proteins suspected to be involved in autophagy.While this unique approach provides a list of potential proteins involved in sperm mitochondria clearance it's (only) a starting point for many future studies and does not provide the demonstration that any of these proteins has indeed a role in the processes leading to sperm mitochondria clearance since the protein identified may also be involved in other processes going-on in the oocyte at this time of early development.

We thank reviewer 1 for positive comments. We added a sentence in Discussion addressing the obvious shortcoming of present study, as further functional validations of candidate mitophagy factors are planned.

Concerning the localisation of the 6 proteins further analysed, the authors must add how much the presented picture represents the observed patterns. They must include the details on the fraction of spermatozoa and embryos displaying the presented pattern.

We now specify that the patterns depicted in manuscript are typical and representative of data from at least three replicates of immunolabeling in spermatozoa and zygotes. For each of these replicates, 200 spermatozoa were examined per replicate of the cell-free system co-incubation or 20 zygotes per replicate. The displayed patterns were observed between 65-88% in examined spermatozoa/zygotes. Invariably, the signal displayed in manuscript is the typical pattern that was seen in a majority of cells. This information has now been added to the Materials & Methods section for clarification.

**Reviewer #2 (Public Review):**
Mitochondria are essential cellular organelles that generate ATPs as the energy source for maintaining regular cellular functions. However, the degradation of sperm-borne mitochondria after fertilization is a conserved event known as mitophagy to ensure the exclusively maternal inheritance of the mitochondrial DNA genome. Defects on post-fertilization sperm mitophagy will lead to fatal consequences in patients. Therefore, understanding the cellular and molecular regulation of the postfertilization sperm mitophagy process is critically important. In this study, Zuidema et. al applied mass spectrometry in conjunction with a porcine cell-free system to identify potential autophagic cofactors involved in post-fertilization sperm mitophagy. They identified a list of 185 proteins that might be candidates for mitophagy determinants (or their co-factors). Despite the fact that 6 (out of 185) proteins were further studied, based on their known functions, using a porcine cell-free system in conjunction with immunocytochemistry and Western blotting, to characterize the localization and modification changes these proteins, no further functional validation experiments were performed. Nevertheless, the data presented in the current study is of great interest and could be important for future studies in this field.

We thank reviewer 2 for positive comments. As we explain in our response to Editors and Reviewer 1, further validation studies will be resumed once the availability of slaughterhouse ovaries for such studies improves. Examples of such functional validation of pro-mitophagic proteins SQSTM1 andVCP are included in our previous studies (DOI: 10.1073/pnas.1605844113 andDOI: 10.3390/cells10092450) that led to the development of cell-free system reported here, and are cited in present study.

**Reviewer #3 (Public Review):**
In this manuscript, a cytosolic extract of porcine oocytes is prepared. To this end, the authors have aspirated follicles from ovaries obtained from by first maturing oocytes to meiose 2 metaphase stage (one polar body) from the slaughterhouse. Cumulus cells (hyaluronidase treatment) and the zona pellucida (pronase treatment) were removed and the resulting naked mature oocytes (1000 per portion) were extracted in a buffer containing divalent cation chelator, beta-mercaptoethanol, protease inhibitors, and a creatine kinase phosphocreatine cocktail for energy regeneration which was subsequently triple frozen/thawed in liquid nitrogen and crushed by 16 kG centrifugation. The supernatant (1.5 mL) was harvested and 10 microliters of it used for interaction with 10,000 permeabilized boar sperm per 10 microliter extract (which thus represents the cytosol fraction of 6.67 oocytes). The sperm were in this assay treated with DTT and lysoPC to prime the sperm's mitochondrial sheath. After incubation and washing these preps were used for Western blot (see point 2) for Fluorescence microscopy and for proteomic identification of proteins.Points for consideration:1. The treatment of sperm cells with DTT and lysoPC will permeabilize sperm cells but will also cause the liberation of soluble proteins as well as proteins that may interact with sperm structures via oxidized cysteine groups (disulfide bridges between proteins that will be reduced by DTT).

This is certainly a possibility, the lysoPC and DTT permeabilization steps were designed to mimic natural processing (plasma membrane removal and sperm protein disulfide bond reduction), which the spermatozoa would undergo during fertilization. However, we do realize that this is a chemically induced processing and thus is not a perfect recapitulation of fertilization processes. However, in this study and in previous studies with this system, we were able to show alignment between proteomic interactions taking place in the cell-free system and within the zygotes.

1. Figure 3: Did the authors really make Western blots with the amount of sperm cells and oocyte extracts as the description in the figures is not clear? This point relates to point 1. The proteins should also be detected in the following preparations (1) for the oocyte extract only (done) (2) for unextracted nude oocytes to see what is lost by the extraction procedure in proteins that may be relevant (not done) (3) for the permeabilized (LPC and DTT treated and washed) sperm only (not done) (4) For sperm that were intact (done) (5) After the assay was 10,000 permeabilized sperm and the equivalent of 6.67 oocyte extracts were incubated and were washed 3 times (or higher amounts after this incubation; not done). Note that the amount of sperm from one assay (10,000) likely will give insufficient protein for proper Western blotting and or Coomassie staining. In the materials and methods, I cannot find how after incubation material was subjected to western blotting the permeabilized sperm. I only see how 50 oocyte extracts and 100 million sperm were processed separately for Western blot.

The authors did make Western blots with the number of spermatozoa and oocytes stated in the materials and methods, a total protein equivalent of 10 to 20 million spermatozoa (equivalent to ~20-40 µg of total protein load) and 100 MII oocytes (equivalent to ~20 µg of total protein load). These numbers have been corrected in the Materials & Methods. Also, we did find in the Materials & Methods section that the Co-Incubation of Permeabilized Mammalian Spermatozoa with Porcine Oocyte Extracts section refers to using cell-free exposed spermatozoa for electrophoresis; however, for none of the presented Western blot work was this true. Rather, all of the presented Western blots as per their descriptions are utilizing ejaculated or capacitated sperm or oocytes. This line has been removed from the Materials & Methods to reduce confusion.

Regarding preparation (2), we have previously assessed the difference between oocyte extract and intact oocytes in this manner internally and we are certainly losing proteins due to the oocyte extraction process. We make caveats in this vein throughout the article such as: “Furthermore, this cell-free system while useful does not perfectly capture all the events which take place during in vivo fertilization. The cell-free system is intended to mimic early fertilization events but is presumably not the exact same as in vitro fertilization.”

1. Figures 4, 5, 6, 7, and 8 see point 2. I do miss beyond these conditions also condition 1 despite the fact that the imaged ooplasm does show positive staining.

For all the presented Western blots, the tissue type is stated in the image description and the protocol which was used to prepare these samples is stated in the Materials & Methods.

1. These points 1-3 are all required for understanding what is lost in the sperm and oocyte treatments prior to the incubation step as well as the putative origin of proteins that were shown to interact with the mitochondrial sheath of the oocyte extract incubated permeabilized sperm cells after triple washing. Is the origin from sperm only (Figs 5-8) or also from the oocyte? Is the sperm treatment prior to incubation losing factors of interest (denaturation by DTT or dissolving of interacting proteins preincubation Figs 3-8)?

The authors understand that there are proteins and interactions lost on both sides of the cellfree system equation and we have added a sentence to the Discussion to caveat this limitation in the system.

1. Mass spectrometry of the permeabilized sperm incubated with oocyte extracts and subsequent washing has been chosen to identify proteins involved in the autophagy (or cofactors thereof). The interaction of a number of such factors with the mitochondrial sheath of sperm has been shown in some cases from sperm and others for an oocyte origin. Therefore, it is surprising that the authors have not sub-fractionated the sperm after this incubation to work with a mitochondrial-enriched subfraction. I am very positive about the porcine cell-free assay approach and the results presented here. However, I feel that the shortcomings of the assay are not well discussed (see points 1-5) and some of these points could easily be experimentally implemented in a revised version of this manuscript while others should at least be discussed.

We agree that the use of a mitochondrial-enriched subfraction for further analysis would be interesting and useful. We are actively developing experimental protocols for oocyte extract coincubation with isolated sperm heads and tails, and eventually with purified mitochondrial sheaths. However, such experiments are contingent upon our access to porcine oocytes, which has continued to be a struggle since the COVID-19 pandemic compromised our ability to attain oocytes in large, cheap, and reliable quantities. This was a continuous problem with preparing materials for this very paper and has continued to be an issue for our laboratory as well as many others at our university and across the country. We continue to maximize oocytes every time we can get access to them, but the unfortunate reality is that this access has become sparce and unreliable over the past three years.